# Domain-Adapted Diffusion Model for PROTAC Linker Design Through the Lens of Density Ratio in Chemical Space

**Zixing Song** [1]  **Ziqiao Meng** [2]  **José Miguel Hernández-Lobato** [1]

## Abstract

Proteolysis-targeting chimeras (PROTACs) are a groundbreaking technology for targeted protein degradation, but designing effective linkers that connect two molecular fragments to form a drug-candidate PROTAC molecule remains a key challenge. While diffusion models show promise in molecular generation, current diffusion models for PROTAC linker design are typically trained on small molecule datasets, introducing distribution mismatches in the chemical space between small molecules and target PROTACs. Direct fine-tuning on limited PROTAC datasets often results in overfitting and poor generalization. In this work, we propose DAD-PROTAC, a domain-adapted diffusion model for PROTAC linker design, which addresses this distribution mismatch in chemical space through density ratio estimation to bridge the gap between small-molecule and PROTAC domains. By decomposing the target score estimator into a pre-trained score function and a lightweight score correction term, DAD-PROTAC achieves efficient fine-tuning without full retraining. Experimental results demonstrate its superior ability to generate high-quality PROTAC linkers.

## 1. Introduction

Proteolysis-targeting chimera (PROTAC) (Gharbi & Mercado, 2024) has emerged as a useful technology for targeted protein degradation (TPD) (Zhao et al., 2022) in drug discovery (Schapira et al., 2019), leveraging the ubiquitin-proteasome system to remove specific unwanted disease-relevant proteins. A PROTAC is a hetero-bifunctional molecule consisting of three components: a ligand that

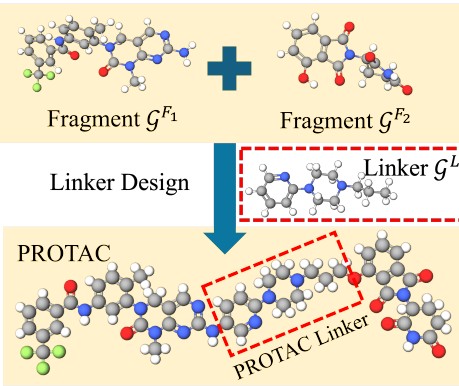

Figure 1: PROTAC linker design. Given two molecule fragments, the goal is to generate a linker that connects them to form a PROTAC molecule (Details in Figure 6).

binds to the target protein for degradation, another ligand that binds to an E3 ligase, and a linker that connects two ligands (Bemis et al., 2021). Unlike traditional small molecule drugs that only temporarily suppress protein function (Martín-Acosta & Xiao, 2021), PROTACs achieve complete protein removal with enhanced selectivity and reduced off-target effects. However, the design of effective PRO-TACs faces significant challenges, particularly in optimizing the linker structure (Troup et al., 2020). The PROTAC linker must ensure precise spatial arrangement of atoms in 3D space, with accurate atom and bond types, while simultaneously meeting complex physicochemical constraints when it is connected to the two ligands (Figure 1).

Recent advances in diffusion models have shown remarkable success in molecule generation tasks (Hoogeboom et al., 2022; Xu et al., 2023; Huang et al., 2024), making them powerful tools for PROTAC linker design (Igashov et al., 2024; Guan et al., 2023; Li et al., 2024). In general, these diffusion models iteratively transform random noise into valid linker structures through a learned denoising process, while preserving chemical and geometric constraints (Morehead & Cheng, 2024). Hence, they can learn to generate new samples that closely match the distribution of the linker in the given training dataset. Current approaches predominantly train these diffusion models on datasets con-

[1]Department of Engineering, University of Cambridge, Cambridge, United Kingdom [2]National University of Singapore, Singapore. Correspondence to: Zixing Song <zs456@cam.ac.uk>.

*Proceedings of the 42$^{nd}$ International Conference on Machine Learning*, Vancouver, Canada. PMLR 267, 2025. Copyright 2025 by the author(s).

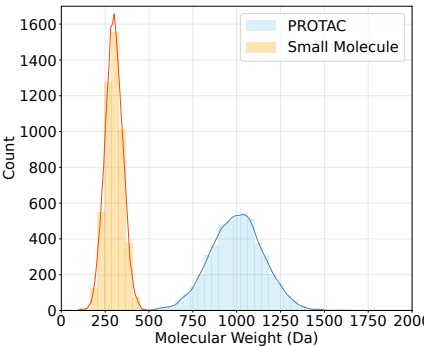

Figure 2: The distributions of molecular weight in PRO-TACs versus small molecules in the chemical space.

structed from the readily available *small molecule* domain, such as ZINC (Irwin & Shoichet, 2005), where existing molecules are artificially split into two fragments connected by a linker (Igashov et al., 2024). This simple training setup enables the models to capture the relationships between the resulting fragments and the linker, establishing a foundation for generating the linker based on the given fragments from the small molecule domain only.

Therefore, a critical challenge in applying diffusion models to the PROTAC linker design task lies in the distribution mismatch *in the chemical space* between the *small molecule* domain, where the models are typically trained, and the *PROTAC* domain, where they are ultimately deployed. Ideally, training datasets would be derived directly from the PROTAC domain, such as experimentally validated ternary complexes in the PROTAC-DB dataset (Weng et al., 2021). However, because synthesizing PROTACs is expensive and time-consuming, the PROTAC-DB dataset is much smaller than small molecule datasets like ZINC. As illustrated in Figure 2, the chemical space of PROTACs significantly differs from that of small molecules, particularly in molecular weight, reflecting the larger size of PROTACs. More discussions on the differences between the two domains can be found in Appendix B.1. The distribution mismatch in the chemical space introduces biases to the generation of PRO-TAC linkers by diffusion models trained on small molecules, limiting their ability to capture PROTAC-specific properties. While direct fine-tuning on PROTAC datasets seems like a natural solution, the limited size of such datasets often leads to overfitting and poor generalization empirically.

This distribution mismatch highlights the need for a domain-adapted diffusion model that explicitly incorporates the ratio of probability density between the small molecule and PRO-TAC domains in the chemical space. Modeling this density ratio enhances the fine-tuning process of diffusion models for general linker design, improves alignment with the PRO-TAC domain, and enables specific PROTAC linker design.

In this work, we introduce DAD-PROTAC, a **D**omain-**A**dapted **D**iffusion model for **PROTAC** linker generation that effectively transfers knowledge from the small molecule domain to the PROTAC domain. DAD-PROTAC is pre-trained on small molecule datasets with a Gaussian diffusion process for continuous features (atom coordinates) and a multinomial diffusion process for discrete features (atom/bond types). The key innovation lies in our fine-tuning strategy, which leverages density ratio estimation in chemical space to perform domain adaptation with theoretical guidance. Instead of directly retraining the model on the PROTAC dataset, we decompose the optimal score estimator for the fine-tuning phase into two components: a pre-trained score function from the small molecule domain and a score correction term based on the density ratio between domains. This approach not only highlights the role of density ratios in guiding fine-tuning but also improves efficiency by learning the score correction term as a lightweight classifier rather than retraining the entire model.

Our contributions can be summarized as follows.

- We propose a novel domain-adapted diffusion model (DAD-PROTAC) for PROTAC linker design that explicitly utilizes density ratio in chemical space between the small molecule domain and the PROTAC domain.

- We theoretically show that the score estimator for the target PROTAC domain can be decomposed into the pre-trained score function and another score correction term involved with density ratio estimation.

- We present an efficient fine-tuning approach by learning to approximate the score correction term guided by the density ratio, avoiding full model re-training.

## 2. Method

### 2.1. Problem Definition

Following existing works (Guan et al., 2023), we represent each molecular fragment as a 3D graph $\mathcal{G}^F = \{\mathcal{V}^F, \mathcal{E}^F\}$. The atom set $\mathcal{V}^F$ and the atom bond set $\mathcal{E}^F$ are associated with the one-hot atom type features $A^F \in \mathbb{R}^{N_f \times N_a}$ and the one-hot bond type features $B^F \in \mathbb{R}^{N_f \times N_f \times N_b}$, respectively. Here, $N_f, N_a, N_b$ denote the number of atoms in the fragment, the number of atom types, and the number of bond types. The atom coordinates $X^F \in \mathbb{R}^{3 \times N_f}$ of the fragments are in the 3D space. Hence, each molecular fragment $\mathcal{G}^F$ is compactly represented by a triple $\{A^F, B^F, X^F\}$.

Similarly, the linker $\mathcal{G}^L = \{\mathcal{V}^L, \mathcal{E}^L\}$ can also be represented as $\{A^L, B^L, X^L\}$. Here, $A^L \in \mathbb{R}^{N_l \times N_a}$ denotes the atom type features of the linker. $B^L \in \mathbb{R}^{N_l \times N \times N_b}$ denotes the bond type features of the linker. Here, $N = N_{f_1} + N_{f_2} + N$ is the total number of atoms of two frag-

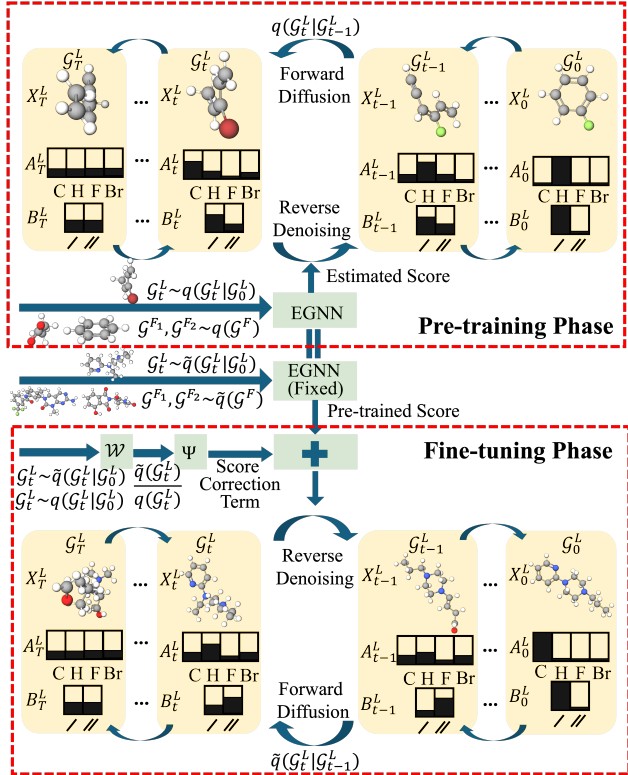

Figure 3: The proposed DAD-PROTAC model. In the pre-training phase, DAD-PROTAC performs a joint diffusion process for the three components of the linker from the small molecule domain $q(\mathcal{G}_t^L)$ and utilizes the EGNN model to estimate the scores for the reverse denoising process. In the fine-tuning phase, DAD-PROTAC obtains the base estimated scores from the pre-trained model with new samples from the PROTAC domain $\tilde{q}(\mathcal{G}_t^L)$ as the input and learns to approximate the score correction term via two lightweight neural networks $\mathcal{W}$ and $\Psi$ under the guidance of the density ratio $\frac{\tilde{q}(\mathcal{G}_t^L)}{q(\mathcal{G}_t^L)}$ in the chemical space between two domains.

ments and the linker. Note that the edge set of the linker contains both the bonds within the linker itself and its bonds connected with the two fragments. $X^L \in \mathbb{R}^{3 \times N_L}$ denotes the 3D coordinates of the atoms in the linker.

Given a pre-trained parameterized generative model $p_\theta(\mathcal{G}^L \mid \mathcal{G}^{F_1}, \mathcal{G}^{F_2})$ for the linker distribution $q(\mathcal{G}^L)$ in the source small molecule domain, we aim to fine-tune this model for the shifted linker distribution $\tilde{q}(\mathcal{G}^L)$ in the target PROTAC domain. We resort to diffusion models (Hoogeboom et al., 2022; Schneuing et al., 2024) as the backbone. The related work is extensively discussed in Appendix A.

## 2.2. Pre-training in the Small Molecule Domain

In the pre-training phase (Figure 3), the diffusion model is trained on a large-scale, readily accessible dataset.

The dataset is constructed using matched-molecular pairs (MMPs) to break molecules into fragment molecule triplets $\{\mathcal{G}^L, \mathcal{G}^{F_1}, \mathcal{G}^{F_2}\}$ in the small molecule domain.

A diffusion probabilistic model consists of two interrelated Markov chains: a forward diffusion process and a reverse generative process (Ho et al., 2020). The forward diffusion process gradually adds noise to the linker data $\mathcal{G}^L$, transforming it into a pure noise distribution. In contrast, the reverse generative process is designed to learn how to take that noise-perturbed data and gradually remove the noise, generating new data samples that resemble the original distribution $q(\mathcal{G}^L \mid \mathcal{G}^{F_1}, \mathcal{G}^{F_2})$ in the source domain for small molecules. Based on the representation of the linker in Section 2.1, the distribution $q(\mathcal{G}^L \mid \mathcal{G}^{F_1}, \mathcal{G}^{F_2})$ can be further decomposed as a product of atom coordinates in the linker $q(X^L \mid \mathcal{G}^{F_1}, \mathcal{G}^{F_2})$ and the linker's atom/bond types $q(A^L, B^L \mid \mathcal{G}^{F_1}, \mathcal{G}^{F_2})$. We will discuss how these diffusion processes are constructed during the pre-training phase.

### 2.2.1. DIFFUSION ON ATOM COORDINATES $X^L$

The diffusion process on $X^L$ utilizes the standard Gaussian diffusion kernels and thus follows the well-established Denoising diffusion probabilistic models (DDPMs) (Ho et al., 2020). At each time step $t$, a small Gaussian noise is introduced for $X_t^L$ based on the previous step $X_{t-1}^L$ with fixed variance schedules $\beta_1, \cdots, \beta_t$ as in Equation (1).

$$q(X_t^L \mid X_{t-1}^L) = \mathcal{N}(X_t^L; \sqrt{1 - \beta_t} X_{t-1}^L, \beta_t \mathbf{I}). \quad (1)$$

This forward process admits sampling $X_t^L$ at an arbitrary time step $t$ directly based on the initial given clean data sample $X_0^L$ with a closed-form solution as in Equation (2).

$$q(X_t^L \mid X_0^L) = \mathcal{N}(X_t^L; \sqrt{\bar{\alpha}_t} X_0^L, (1 - \bar{\alpha}_t) \mathbf{I}). \quad (2)$$

Here, $\alpha_t = 1 - \beta_t$, $\bar{\alpha}_t = \Pi_{s=1}^t \alpha_s$. Based on the reparameterization trick, we have $X_t^L = \sqrt{\bar{\alpha}_t} X_0^L + \sqrt{1 - \bar{\alpha}_t} \epsilon$ for $\epsilon \sim \mathcal{N}(0, \mathbf{I})$. The posterior distribution is obtained as

$$q(X_{t-1}^L \mid X_t^L, X_0^L) = \mathcal{N}(X_{t-1}^L; \tilde{\mu}_t(X_t^L, X_0^L), \tilde{\beta}_t \mathbf{I}). \quad (3)$$

Here, $\tilde{\mu}_t(X_t^L, X_0^L) = \frac{\sqrt{\bar{\alpha}_{t-1}} \beta_t}{1 - \bar{\alpha}_t} X_0^L + \frac{\sqrt{\alpha_t}(1 - \bar{\alpha}_{t-1})}{1 - \bar{\alpha}_t} X_t^L$, and $\tilde{\beta}_t = \frac{1 - \bar{\alpha}_{t-1}}{1 - \bar{\alpha}_t} \beta_t$.

The reverse generative denoising process on $X^L$ aims to invert the diffusion trajectory by approximating the original linker's coordinates $X_0^L$ based on the noisy $X_t^L$ at the time step $t$ using a neural network with parameters $\theta$. It is more effective to predict the Gaussian noise $\epsilon$ with the loss as

$$\mathcal{L}^X = \mathbb{E}_t \mathbb{E}_{X_0^L} \mathbb{E}_\epsilon \left[ \| \epsilon_\theta(\sqrt{\bar{\alpha}_t} X_0^L + \sqrt{1 - \bar{\alpha}_t} \epsilon, t) - \epsilon \|_2^2 \right]. \quad (4)$$

The dynamics of predicting $\epsilon_\theta(X_t^L, t)$ will later be elaborated in Section 2.2.3. If we let $s_\theta(X_t^L) = -\frac{\epsilon_\theta(X_t^L, t)}{\sqrt{1 - \bar{\alpha}_t}}$ and

note that the score of posterior $\nabla_{X_t^L} \log q(X_t^L \mid X_0^L) = -\frac{\epsilon}{\sqrt{1-\bar{\alpha}_t}}$, we could further simplify Equation (4) as,

$$\mathcal{L}^X = \mathbb{E}_t \mathbb{E}_{X_0^L} \mathbb{E}_{X_t^L \mid X_0^L} \| s_\theta(X_t^L, t) - \nabla_{X_t^L} \log q(X_t^L \mid X_0^L) \|_2^2. \tag{5}$$

### 2.2.2. DIFFUSION ON ATOM(BOND) TYPES $A^L(B^L)$

The diffusion process of the atom or bond types of the linker is similar to the one for the atom coordinates, but now we resort to multinomial diffusion (Hoogeboom et al., 2021). This is because unlike the atom coordinates data which lie in the continuous space, the atom (bond) type features of the linker are one-hot vectors as the discrete categorical data. We use atom types $A^L$ as an example from now on. The case for bond types $B^L$ follows the same manner.

We use the categorical distribution for the forward process that has a $\beta_t$ chance of resampling a category uniformly.

$$q(A_t^L \mid A_{t-1}^L) = \text{Cat} \left( A_t^L; (1 - \beta_t) A_{t-1}^L + \beta_t / N_a \right). \tag{6}$$

$N_a$ is the total number of atom types. Due to the property of the Markov chain, the probability of any $A_t^L$ given $A_0^L$ can be easily obtained in Equation (7).

$$q(A_t^L \mid A_0^L) = \text{Cat} \left( A_t^L; \bar{\alpha}_t A_0^L + (1 - \bar{\alpha}_t) / N_a \right). \tag{7}$$

Here, we also have $\alpha_t = 1 - \beta_t$, $\bar{\alpha}_t = \Pi_{s=1}^t \alpha_s$. The categorical posterior can be computed in the closed form as

$$q(A_{t-1}^L \mid A_t^L, A_0^L) = \text{Cat}(A_{t-1}^L; \Pi / \sum_{k=1}^{N_a} \Pi_k). \tag{8}$$

$\Pi = [\alpha_t A_t^L + (1 - \alpha_t) / N_a] \odot [\bar{\alpha}_{t-1} A_0^L + (1 - \bar{\alpha}_{t-1}) / N_a]$. To overcome the difficulty in predicting the noise for discrete atom type data, we utilize the denoising concrete score matching method (Meng et al., 2024). To generalize the score for the discrete setting of $A_t^L$, the concrete score of any distribution $p(x)$ at data point $x$ is $c_{p(x)}$ and is defined by the rate of change of the probabilities w.r.t. local directional changes of the input $x$ within a neighborhood of $\mathcal{N}(x) = \{x_{n_i}\}_{i=1}^k$, i.e., $c_{p(x)} = [\frac{p(x_{n_1}) - p(x)}{p(x)}, \cdots, \frac{p(x_{n_k}) - p(x)}{p(x)}]^\top$. Similar to Equation (4), the training loss now becomes

$$\mathcal{L}^A = \mathbb{E}_t \mathbb{E}_{A_0^L} \mathbb{E}_{A_t^L \mid A_0^L} \| c_\theta(A_t^L, t) - c_{q(A_t^L \mid A_0^L)}(A_t^L) \|_2^2. \tag{9}$$

The dynamics of predicting $c_\theta(A_t^L)$ will be elaborated next.

### 2.2.3. EQUIVARIANT GRAPH NEURAL NETWORKS FOR DENOISING OVER THE PERTURBED LINKER $\mathcal{G}_t^L$

To incorporate the information from fragments $\mathcal{G}^{F_1}, \mathcal{G}^{F_2}$ for denoising $\mathcal{G}_t^L$, we use the whole graph $\mathcal{G}^P = \{\mathcal{V}^P, \mathcal{E}^P\}$ as the input with $\mathcal{V}^P = \mathcal{V}^{F_1} \cup \mathcal{V}^{F_2} \cup \mathcal{V}^L$. We require that the atom embeddings $\mathbf{h}_i$ $(i \in \mathcal{V}^P)$ and bond embeddings $\mathbf{e}_{ij}$ $(i, j \in \mathcal{V}^P)$ output by a neural network

$\phi_\theta$ are both invariant to global SE(3)-transformation. Furthermore, atom coordinates of the linker $\mathbf{x}_i$ $(i \in \mathcal{V}^L)$ updated by $\phi_\theta$ should be SE(3)-equivariant. The proposed DAD-PROTAC model learns to predict both $s_\theta(X_t^L, t)$ and $c_\theta(A_t^L, t)$ with equivariance constraints, utilizing an $L$-th layer E(n) Equivariant Graph Neural Network (EGNN) model $\phi_\theta$ (Satorras et al., 2021), as $[s_\theta, c_\theta] = \phi_\theta(\mathcal{G}^{P_t}, t) = \phi_\theta([A_t^L, B_t^L, X_t^L], \mathcal{G}^{F_1}, \mathcal{G}^{F_2}, t)$. Hence, they are updated in the $l$-th EGNN layer as follows.

$$\tilde{\mathbf{e}}_{ij} = \phi_d \left( \mathbf{e}_{ij}^l, \| \mathbf{x}_i^l - \mathbf{x}_j^l \|_2^2 \right)$$

$$\mathbf{h}_i^{l+1} = \mathbf{h}_i^l + \sum_{j \in \mathcal{V}^P \setminus \{i\}} \phi_h \left( \mathbf{h}_i^l, \mathbf{h}_j^l, \tilde{\mathbf{e}}_{ij}, t \right)$$

$$\mathbf{e}_{ij}^{l+1} = \mathbf{e}_{ij}^l + \sum_{k \in \mathcal{V}^P \setminus \{i\}} \phi_h \left( \mathbf{h}_k^l, \mathbf{h}_i^l, \tilde{\mathbf{e}}_{ki}, t \right) + \sum_{k \in \mathcal{V}^P \setminus \{j\}} \phi_h \left( \mathbf{h}_j^l, \mathbf{h}_k^l, \tilde{\mathbf{e}}_{jk}, t \right)$$

$$\mathbf{x}_i^{l+1} = \mathbf{x}_i^l + \sum_{j \in \mathcal{V}^P \setminus \{i\}} \left( \mathbf{x}_i^l - \mathbf{x}_j^l \right) \phi_x \left( \mathbf{h}_i^{l+1}, \mathbf{h}_j^{l+1}, \mathbf{e}_{ij}^{l+1}, t \right) \cdot \mathbb{1}_{\text{linker}}$$

Here, $\mathbb{1}_{\text{linker}}$ is the linker atom mask, a binary vector where each element indicates whether an atom belongs to the linker region or not. The initial atom and bond embeddings are encoded with simple multilayer perceptions (MLP) based on the corresponding atom and bond types as $\mathbf{h}_i^0 = \text{MLP}(A_t^L[i])$ and $\mathbf{e}_{ij}^0 = \text{MLP}(B_t^L[ij])$, respectively. $A_t^L[i]$ and $B_t^L[ij]$ are the type of atom $i$ and the type of bond $ij$ in the noise-perturbed linker $\mathcal{G}_t^L$, respectively. The EGNN model $\phi_\theta$ consists of three sets of parameters $\phi_d$, $\phi_h$, and $\phi_x$, which can all be instantiated with MLPs. The final embeddings $\{\mathbf{h}_i^L\}$ and $\{\mathbf{e}_{ij}^L\}$ with $i, j \in \mathcal{V}^L$ will also be fed into another two MLPs to obtain the predicted concrete scores $c_\theta(A_t^L, t) = \text{MLP}(\{\mathbf{h}_i^L\})$ in Equation (9), and $c_\theta(B_t^L, t) = \text{MLP}(\{\mathbf{e}_{ij}^L\})$. Similarly, we initialize $\mathbf{x}_i^0 = X_t^L[i]$ and set the score estimator $s_\theta(X_t^L, t) = \{\mathbf{X}_i^L\}$.

### 2.3. Fine-tuning in the PROTAC Domain

During the fine-tuning phase (Figure 3), the proposed DAD-PROTAC model transfers knowledge from the pre-trained diffusion model in the small molecule domain, and learn how to further generate samples aligned with the target distribution in the PROTAC domain. A straightforward approach is to directly retrain the model using the pre-trained weights as initialization. However, this naive method often leads to overfitting issues (Lutati & Wolf, 2023), as it tends to capture noise from the limited PROTAC training data samples rather than learning high-level, domain-specific features critical for PROTAC design (Gharbi & Mercado, 2024). Full fine-tuning of all parameters in $\phi_\theta$ further incurs significant computational overhead and prolonged convergence times (Xie et al., 2023). Freezing early layers of the EGNN denoising network can reduce computational costs and mitigate overfitting (Moon et al., 2022), but it still cannot explicitly learn high-level, domain-specific features necessary for better adaptability in the PROTAC linker design with theoretical guarantees.

### 2.3.1. CONNECTION WITH DENSITY RATIO IN THE CHEMICAL SPACE

The fine-tuning process of the proposed DAD-PROTAC model is grounded in its theoretical connection to the density ratio between PROTACs and small molecules in the chemical space, as shown in Theorem 2.1. Specifically, motivated by the previous work (Ouyang et al., 2024; Kim et al., 2024), we can show that the optimal score estimator for the target PROTAC domain during fine-tuning differs from the pre-trained score for the small molecule domain by a correction term that accounts for the density ratio between these two domains in the chemical space. This correction term ensures the model adapts effectively to the distinct characteristics of PROTAC linkers.

**Theorem 2.1.** *Let $q(\mathcal{G}^L)$ and $\tilde{q}(\mathcal{G}^L)$ denote the distributions of linkers in the chemical space of the small molecule domain and the PROTAC domain, respectively. Assume the forward diffusion process on both domains is identical, i.e.,*

$$q(\mathcal{G}_t^L \mid \mathcal{G}_0^L) = \tilde{q}(\mathcal{G}_t^L \mid \mathcal{G}_0^L). \tag{10}$$

*The score estimator $s_{\tilde{\phi}^*}(X_t^L, t)$ for the target PROTAC domain during the fine-tuning phase can be decomposed as*

$$s_{\tilde{\phi}^*}(X_t^L, t) = \underbrace{\nabla_{X_t^L} \log q(X_t^L)}_{\text{pre-trained score function}} + \underbrace{\nabla_{X_t^L} \log \mathbb{E}_{q(X_0^L \mid X_t^L)} \frac{\tilde{q}(X_0^L)}{q(X_0^L)}}_{\text{score correction}}. \tag{11}$$

*Similarly, the concrete score estimator $c_{\tilde{\phi}^*}(A_t^L, t)$ for the target PROTAC domain can also be decomposed as*

$$c_{\tilde{\phi}^*}(A_t^L, t) = \underbrace{c_{q(A_t^L)}(A_t^L)}_{\substack{\text{pre-trained concrete} \\ \text{score function}}} + \underbrace{c_{q(A_t^L)}\left( \mathbb{E}_{q(A_0^L \mid A_t^L)} \frac{\tilde{q}(A_0^L)}{q(A_0^L)} \right)}_{\text{concrete score correction}}. \tag{12}$$

Theorem 2.1 highlights the critical role of the density ratio $\frac{\tilde{q}(\mathcal{G}^L)}{q(\mathcal{G}^L)}$ in chemical space for effectively adapting the pre-trained model to PROTAC linker design. This density ratio serves as a correction term, $\nabla_{X_t^L} \log \mathbb{E}_{q(X_0^L \mid X_t^L)} \frac{\tilde{q}(X_0^L)}{q(X_0^L)}$ (or $c_{q(A_t^L)}(\mathbb{E}_{q(A_0^L \mid A_t^L)} \frac{\tilde{q}(A_0^L)}{q(A_0^L)})$), allowing the proposed DAD-PROTAC model to refine the score function $\nabla_{X_t^L} \log q(X_t^L)$ (or $c_{q(A_t^L)}(A_t^L)$) by the pre-trained model. This adjustment ensures that the model accurately captures the distributional differences between the two domains.

Decomposing the score estimator into two terms in Theorem 2.1 offers two key advantages. For one thing, unlike conventional fine-tuning methods, our approach eliminates the need to directly fine-tune the pre-trained model on the target PROTAC domain. The score estimator is already learned during the pre-training phase, and we now only need to feed the new PROTAC training samples to obtain $\nabla_{X_t^L} \log q(X_t^L)$ (or $c_{q(A_t^L)}(A_t^L)$) as the fixed base score estimation. We instead shift the computational overhead to train

**Algorithm 1** Fine-tuning Phase of DAD-PROTAC

---

**input** Samples from source small molecule domain $q(\mathcal{G}^L)$ and target PROTAC domain $\tilde{q}(\mathcal{G}^L)$, pre-trained (fine-tuning) diffusion process $q(\mathcal{G}_t^L \mid \mathcal{G}_0^L) = \tilde{q}(\mathcal{G}_t^L \mid \mathcal{G}_0^L)$

/* Estimate the density ratio via a binary classifier $\mathcal{W}$ first */

1: **repeat**
2:      Sample small molecule data $\mathcal{G}_0^L \sim q(\mathcal{G}^L)$ with size of $m$, and PROTAC data $\mathcal{G}_0^L \sim \tilde{q}(\mathcal{G}^L)$ with size of $n$.
3:      Sample $t \sim \mathcal{U}(1, T)$ and perturb the sampled data via diffusion process $q(\mathcal{G}_t^L \mid \mathcal{G}_0^L) = \tilde{q}(\mathcal{G}_t^L \mid \mathcal{G}_0^L)$.
4:      Perform gradient descent step over the binary classifier $\mathcal{W}$ via the losses in Equations (16), (23) and (24).
5: **until** converged.
6: Set the density ratio in chemical space term appeared in Equations (13), (19) and (20) as $\frac{\tilde{q}(\mathcal{G}_t^L)}{q(\mathcal{G}_t^L)} = \frac{1 - \mathcal{W}(\mathcal{G}_t^L, t)}{\mathcal{W}(\mathcal{G}_t^L, t)}$.

/* Learn the score correction term for the fine-tuning phase */

7: **repeat**
8:      Sample PROTAC data from $\mathcal{G}_0^L \sim \tilde{q}(\mathcal{G}^L)$ and perturb $\mathcal{G}_0^L$ with noise via diffusion process $\tilde{q}(\mathcal{G}_t^L \mid \mathcal{G}_0^L)$.
9:      Perform gradient descent step over $\psi(\mathcal{G}_t^L, t)$ via the losses in Equations (13), (19) and (20).
10: **until** converged
11: **return** Score estimator for the target PROTAC domain via Equations (15), (21) and (22).

---

an additional neural network to approximate the density ratio term. This term, essentially a classifier as discussed later, substantially reduces computational costs compared to full fine-tuning on the PROTAC domain. For another, the density correction term explicitly incorporates the density ratio between PROTACs and small molecules in the chemical space. By leveraging this density ratio, our method optimally fine-tunes the pre-trained model while highlighting the high-level distribution differences between the two domains. Therefore, our approach enhances model adaptability for PROTAC linker design with theoretical guidance.

### 2.3.2. LEARNING THE SCORE CORRECTION TERM WITH DENSITY RATIO ESTIMATION

We then aim to compute the (concrete) score correction terms in Equation (11) and in Equation (12). We focus on the case of $s_{\tilde{\phi}^*}(X_t^L, t)$ in this section and the case of $c_{\tilde{\phi}^*}(A_t^L, t)$ follows the same idea discussed in Appendix B.4. Assume the density ratio $\frac{\tilde{q}(X_0^L)}{q(X_0^L)}$ is known, we can obtain the expectation $\mathbb{E}_{q(X_0^L \mid X_t^L)} \frac{\tilde{q}(X_0^L)}{q(X_0^L)}$ using Monte Carlo simulation, but it is hard to directly sample from $q(X_0^L \mid X_t^L)$ in the source small molecule domain during fine-tuning. Therefore, following the previous work (Ouyang et al., 2024; Lu et al., 2023), we also build its equivalence to another score correction term that is easier to sample in the target PROTAC domain $\tilde{q}(\mathcal{G}^L)$.

**Theorem 2.2.** *If a parameterized neural network model $\psi(X_t^L, t)$ takes the noise-perturbed $X_t^L$ in the target PRO-TAC domain as the input and $\psi(X_t^L, t)$ is trained with the following loss as,*

$$\mathcal{L}^{\psi_X} = \mathbb{E}_{\tilde{q}(X_0^L, X_t^L)} \left\| \psi(X_t^L, t) - \frac{\tilde{q}(X_t^L)}{q(X_t^L)} \right\|_2^2, \quad (13)$$

*then we have*

$$\psi^*(X_t^L, t) = \arg\min \mathcal{L}^{\psi_X} = \mathbb{E}_{q(X_0^L | X_t^L)} \frac{\tilde{q}(X_0^L)}{q(X_0^L)}. \quad (14)$$

From Theorem 2.2, we now build another neural network $\psi(X_t^L, t)$ to estimate $\mathbb{E}_{q(X_0^L | X_t^L)} \frac{\tilde{q}(X_0^L)}{q(X_0^L)}$ by minimizing the loss function $\mathcal{L}^{\psi_X}$ involved with an easy sampling from the joint distribution of $\tilde{q}(X_0^L, X_t^L)$ in the target PROTAC domain. Instead of using the given limited PROTAC samples only ($\tilde{q}(X_0^L)$ in Equation (11)), we now use the generated noise-perturbed PROTAC samples ($\tilde{q}(X_t^L)$ in Equation (13)) to further improve the estimation of score correction term with more data samples. Therefore, the final score estimator for the target PROTAC domain can be calculated as follows.

$$s_{\tilde{\phi}^*}(X_t^L, t) = \nabla_{X_t^L} \log q(X_t^L) + \nabla_{X_t^L} \log \psi^*(X_t^L, t). \quad (15)$$

To train $\psi(X_t^L, t)$ via the loss in Equation (13), we still need to estimate the density ratio term $\frac{\tilde{q}(X_t^L)}{q(X_t^L)}$ for the marginal distributions perturbed at time $t$ between PROTACs and small molecules in advance. Therefore, we need to train another binary classifier $\mathcal{W}(X_t^L, t)$ beforehand via the cross-entropy loss in Equation (16) to determine whether the perturbed samples are from the source small molecule domain $q(X_t^L)$ or from the target PROTAC domain $q(\tilde{X}_t^L)$.

$$\mathcal{L}^{\mathcal{W}_X} = -\frac{1}{m} \sum_{X_0^L \sim q(X_0^L)} \sum_{X_t^L \sim q(X_t^L | X_0^L)} \log \mathcal{W}(X_t^L, t)$$
$$- \frac{1}{n} \sum_{X_0^L \sim \tilde{q}(X_0^L)} \sum_{X_t^L \sim \tilde{q}(X_t^L | X_0^L)} \log \left(1 - \mathcal{W}(X_t^L, t)\right) \quad (16)$$

The density ratio is approximated as $\frac{\tilde{q}(X_t^L)}{q(X_t^L)} = \frac{1 - \mathcal{W}(X_t^L, t)}{\mathcal{W}(X_t^L, t)}$.

The overall fine-tuning phase of DAD-PROTAC is summarized in Algorithm 1 with more details in Appendix D.2. It comprises two primary steps: density ratio estimation and score function correction, efficiently transferring knowledge from small molecules to PROTACs with theoretical insights. In the density ratio estimation step, a binary classifier $\mathcal{W}$ is trained to quantify the distribution differences in chemical space between small molecules and PROTACs. In the score function correction step, we leverage the neural network $\psi$ to approximate the optimal adjustments needed to adapt the pre-trained score function to the PROTAC domain.

Our fine-tuning approach in the DAD-PROTAC model enjoys two main advantages. First, it is computationally efficient through the score estimator correction rather than full model retraining. Second, it explicitly estimates the density ratio in the chemical space with theoretical rigor for effective domain adaptation. To sum up, the fine-tuning phase of DAD-PROTAC effectively captures the unique characteristics of PROTACs via the lens of density ratio in the chemical space with reduced computational overhead.

## 3. Experiments

### 3.1. Experimental Setup

We discuss the experimental setup for the proposed DAD-PROTAC model with more details in Appendix E.

#### 3.1.1. DATASETS

To evaluate our model, we choose two datasets. One is a subset of ZINC (Irwin & Shoichet, 2005) from the small molecule domain for pre-training and the other one is PROTAC-DB (Weng et al., 2021; 2022) from the PRO-TAC domain for fine-tuning. The size of the PROTAC-DB dataset is significantly small compared to the ZINC dataset.

For the ZINC dataset during the pre-training phase, we follow the previous works (Huang et al., 2022; Igashov et al., 2024; Guan et al., 2023), and obtain the reference conformation for each molecule by running 20 times MMFF optimization (Halgren, 1996) using RDKit (Landrum, 2016) and selecting the one with the lowest energy. Then, these molecules are fragmented by enumerating all double cuts of acyclic single bonds that are not within functional groups. One molecule can therefore result in various combinations of two fragments with a linker between them. The pre-training dataset contains 438,610 samples.

For the PROTAC-DB dataset, which is collected from the literature or calculated by programs, we follow the existing works (Li et al., 2024; Guan et al., 2023). We gather E3 ligands, the warheads (ligands that bind to targets), and the linkers of each PROTAC. We select 365 different warheads as the test set of 327 PROTAC samples, and the remaining as the training set of 2,943 samples for the fine-tuning phase.

#### 3.1.2. BASELINES

For benchmarking, we compare our model with three baselines for the linker design task: 3DLinker (Huang et al., 2022), DiffLinker (Igashov et al., 2024), and LinkerNet (Guan et al., 2023). 3DLinker is the first 3D generative model based on VAE models. DiffLinker is an E(3)-equivariant 3D conditional diffusion model. LinkerNet develops a 3D equivariant diffusion model that jointly generates both fragment poses and the structure of the linker.

Since they do not follow the pretrain-finetuning pipeline, we build their pre-trained counterparts for a fair comparison.

### 3.1.3. EVALUATION METRICS

Similar to molecular generation tasks, We evaluate the generated molecules using both 2D graphs and 3D conformations. We generate 100 PROTAC linker samples per fragment pair in the test set for the evaluation. For 2D metrics, we report standard metrics including validity, uniqueness, novelty (Brown et al., 2019), recovery rate, and property-related metrics including drug-likeness (QED) (Bickerton et al., 2012) and synthetic accessibility (SA) (Ertl & Schuffenhauer, 2009). For 3D conformations, we perform MMFF optimization (Halgren, 1996) and report two metrics: the average minimum energy per fragment pair before optimization ($E_{\min}$), reflecting the overall quality of generated conformations, and the Root Mean Square Deviation (RMSD) of molecular coordinates before and after optimization, representing the deviation from optimal conformations.

### 3.2. Main Experimental Results

For a fair comparison, we evaluate both baseline models trained from scratch on the PROTAC-DB dataset and their pre-trained counterparts. In the latter case, baselines are first pre-trained on the small molecule ZINC dataset and then fine-tuned with limited PROTAC data. The results in Table 1 yield three key observations. First, fine-tuned variants consistently outperform their base models, highlighting the benefits of pre-training on the large-scale ZINC dataset. Second, DAD-PROTAC achieves the best overall performance across most metrics, notably the highest validity and recovery rates. Third, while DiffLinker models perform well in uniqueness and novelty, they often generate chemically invalid structures, making DAD-PROTAC the most practical and robust model overall.

Furthermore, we compare the fine-tuning time of DAD-PROTAC with other methods using direct fine-tuning methods. The relative fine-tuning times, with DAD-PROTAC as the baseline, and the valid rate as the efficacy metric, are summarized in Figure 4(a). DAD-PROTAC achieves superior performance with minimal computational overhead, maintaining the highest validity score. In contrast, traditional fine-tuning methods require significantly more resources, with computational costs ranging from 1.35x to 2.8x that of the proposed DAD-PROTAC model. These results highlight the efficiency of our insightful fine-tuning strategy, which learns to approximate the score correction term under the guidance of density ratio in chemical space, rather than re-training the entire model. In summary, our approach delivers both high performance and exceptional computational efficiency.

### 3.3. Ablation Study

We perform an ablation study on DAD-PROTAC by modifying the fine-tuning components while keeping the pre-training phase unchanged. The results are summarized in Table 2. First, we evaluate standard full fine-tuning, which updates all parameters in the EGNN model to predict the score function. This traditional approach performs poorly due to overfitting. Partial fine-tuning, which updates only the decoder in EGNN, performs slightly better but still yields suboptimal results. Next, we test the direct score correction approximation, where the score correction term is learned via a neural network without using density ratio estimation. This approach achieves the lowest performance across all metrics, revealing that while computationally efficient, it oversimplifies the learning process. These results emphasize that density ratio estimation in the chemical space is critical for accurate score correction. Finally, we analyze the impact of using only fixed clean samples to estimate the density ratio, based on the fact that we use identical forward diffusion processes in both phases. This variant slightly degrades performance compared to DAD-PROTAC's use of noise-perturbed samples, as the latter is trained with a larger and more diverse dataset. Overall, the ablation study demonstrates the importance of each component in DAD-PROTAC, with density ratio estimation playing a pivotal role in achieving optimal performance.

### 3.4. Understanding DAD-PROTAC

To illustrate the impact of fine-tuning in DAD-PROTAC, we evaluate the model's performance on the PROTAC dataset immediately after pre-training without fine-tuning. Results in Figure 4(b) show the distribution of total generated PROTAC linkers according to the number of atoms and the proportion of invalid generated PROTAC structures. Compared to the results of the fine-tuned DAD-PROTAC model in Figure 4(c), fine-tuning clearly yields significant improvements, with remarkably fewer invalid PROTAC structures across all possible linker lengths or the number of atoms in the linker. This is because PROTACs exhibit distinct characteristics compared to traditional small molecules in the pretraining ZINC dataset. We demonstrate that the fine-tuning phase in DAD-PROTAC effectively enhances the model's understanding of the divergence in characteristics of two datasets.

Additionally, Figure 5(a) compares the molecular weight distributions of true PROTACs in the test set with those of generated PROTACs. The high degree of overlap highlights the fine-tuned model's ability to generate PROTACs that closely align with the chemical space of true PROTACs, underscoring the importance of density ratio estimation in the chemical space during fine-tuning.

Table 1: Performance metrics for generated PROTAC linkers on the PROTAC-DB dataset.

| Model | Pre-trained on ZINC | Valid (↑) | Unique (↑) | Novel (↑) | Recover (↑) | QED (↑) | SA (↓) | $E_{\text{min}}$ (↓) | RMSD (↓) |
|---|---|---|---|---|---|---|---|---|---|
| 3DLinker | No | 21.4 ± 0.2 | 25.8 ± 0.3 | 27.3 ± 0.1 | 3.4 ± 0.2 | 0.44 ± 0.2 | 2.98 ± 0.3 | 531.6 ± 12.7 | 1.97 ± 0.3 |
| 3DLinker-Fine-tuning | Yes | 58.6 ± 0.2 | 49.2 ± 0.6 | 56.2 ± 0.5 | 24.3 ± 0.4 | 0.58 ± 0.2 | 2.46 ± 0.1 | 326.4 ± 10.2 | 1.81 ± 0.4 |
| DiffLinker | No | 24.3 ± 0.1 | 98.4 ± 0.1 | 98.5 ± 0.2 | 1.1 ± 0.1 | 0.53 ± 0.3 | 2.58 ± 0.2 | 420.2 ± 13.4 | 2.58 ± 0.1 |
| DiffLinker-Fine-tuning | Yes | 64.0 ± 0.3 | **98.8 ± 0.2** | **99.3 ± 0.2** | 13.6 ± 0.3 | 0.65 ± 0.2 | 2.22 ± 0.4 | 253.0 ± 11.6 | 2.24 ± 0.2 |
| LinkerNet | No | 55.1 ± 0.1 | 46.8 ± 9.3 | 40.4 ± 5.3 | 5.2 ± 0.9 | 0.61 ± 0.1 | 2.35 ± 0.2 | 124.9 ± 6.3 | 1.60 ± 0.3 |
| LinkerNet-Fine-tuning | Yes | 82.9 ± 0.3 | 54.6 ± 7.0 | 63.7 ± 3.8 | 32.8 ± 0.8 | 0.69 ± 0.2 | 1.97 ± 0.1 | 110.7 ± 8.2 | 1.52 ± 0.2 |
| DAD-PROTAC | Yes | **94.8 ± 0.4** | 69.3 ± 0.3 | 71.5 ± 0.3 | **45.7 ± 0.6** | **0.74 ± 0.1** | **1.63 ± 0.1** | **92.4 ± 9.5** | **1.43 ± 0.2** |

Table 2: Ablation study for the proposed DAD-PROTAC model.

| Model | Valid% (↑) | Unique% (↑) | Novel% (↑) | Recover% (↑) | $E_{\text{min}}$ (↓) | RMSD (↓) |
|---|---|---|---|---|---|---|
| DAD-PROTAC w/ standard full fine-tuning | 82.5 ± 0.4 | 57.0 ± 0.2 | 63.1 ± 0.3 | 32.4 ± 0.2 | 117.4 ± 10.4 | 1.57 ± 0.6 |
| DAD-PROTAC w/ standard partial fine-tuning | 83.1 ± 0.6 | 54.4 ± 0.8 | 62.2 ± 0.4 | 34.3 ± 0.5 | 107.0 ± 10.2 | 1.51 ± 0.4 |
| DAD-PROTAC w/ direct score correction approximation | 81.9 ± 0.8 | 49.5 ± 0.4 | 59.7 ± 0.3 | 28.1 ± 0.7 | 203.8 ± 12.3 | 2.27 ± 0.3 |
| DAD-PROTAC w/ density ratio estimation via clean samples | 90.0 ± 4.7 | 62.8 ± 6.1 | 69.3 ± 0.2 | 41.0 ± 0.6 | 105.6 ± 8.9 | 1.49 ± 0.5 |
| DAD-PROTAC | **94.8 ± 0.4** | **69.3 ± 0.3** | **71.5 ± 0.3** | **45.7 ± 0.6** | **92.4 ± 9.5** | **1.43 ± 0.2** |

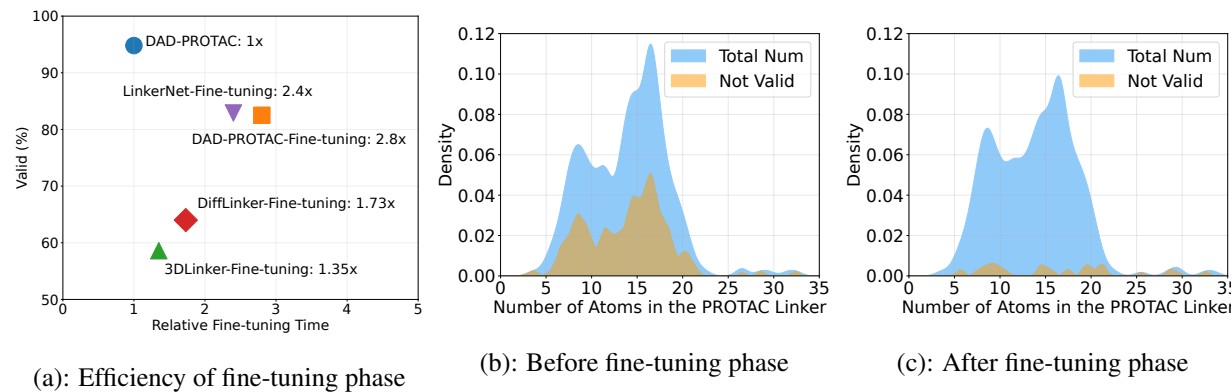

(a): Efficiency of fine-tuning phase

(b): Before fine-tuning phase

(c): After fine-tuning phase

Figure 4: (a): Fine-tuning time comparison. (b)(c): Distribution of generated PROTAC linkers by DAD-PROTAC.

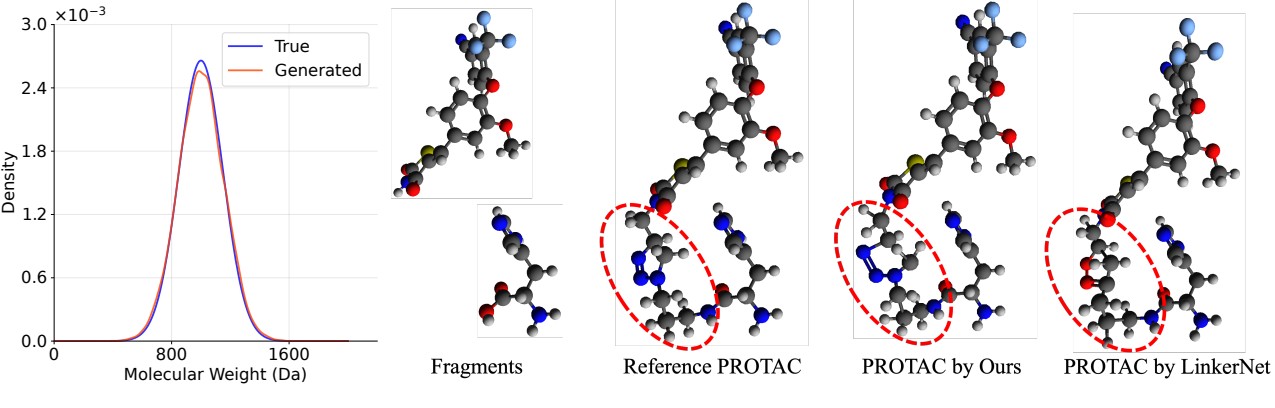

(a): Molecular weight

(b): Visualization of the reference and generated PROTACs given the fragments

Figure 5: (a): Distribution of molecular weight of the test and generated PROTACs. (b): Visualization results.

### 3.5. Visualization

Finally, we visualize examples of PROTAC linker design in Figure 5(b). Given a fragment pair from the test set, we select the best-generated PROTAC from our method and

LinkerNet based on the SCRDKit score. Our model precisely predicts the correct atoms and bonds, greatly matching the reference linker, whereas LinkerNet produces an incorrect structure.

# 4. Conclusion

In this work, we propose DAD-PROTAC, a domain-adapted diffusion model for PROTAC linker design. To address the distribution mismatch between small molecules and PROTACs, DAD-PROTAC decomposes the target score estimator into a pre-trained score function and a lightweight score correction term under the guidance of the density ratio in chemical space, DAD-PROTAC enables efficient fine-tuning, mitigating the overfitting issue.

Despite its strengths, DAD-PROTAC has several limitations. First, it assumes that the number of atoms in the linker is pre-specified, which is unrealistic for real-world applications. Second, the pre-training phase does not consider fragment rotation or protein context. These limitations could potentially be addressed by integrating more advanced diffusion-based models during the pre-training phase.

# Acknowledgements

Zixing Song and José Miguel Hernández-Lobato acknowledge support from EPSRC funding under grant EP/Y028805/1. José Miguel Hernández-Lobato also acknowledges support from a Turing AI Fellowship under grant EP/V023756/1.

# Impact Statement

Our work advances computational methods for PROTAC design, offering potential acceleration in therapeutic development for challenging diseases. This technology could significantly reduce drug development timelines and costs, particularly benefiting areas where traditional small molecule approaches have proven insufficient. However, we acknowledge that powerful molecular generation tools carry inherent risks of potential misuse. To address these concerns, we recommend implementing safeguards including toxicity prediction filters and restricted chemical space exploration using well-characterized building blocks. We advocate for transparent reporting of both capabilities and limitations, and encourage ongoing collaboration between computational scientists, medicinal chemists, and bioethicists to ensure these advances benefit global health while minimizing potential risks.

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

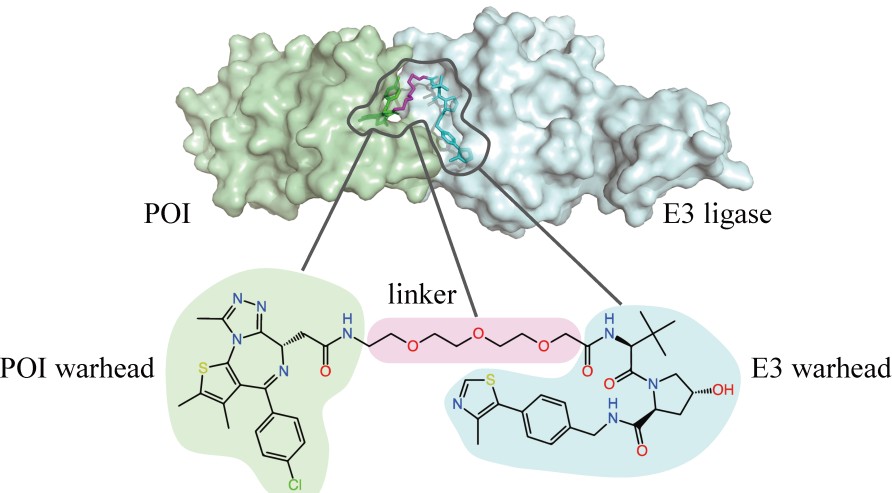

Figure 6: PROTACs are molecules composed of two "warheads" and a connecting linker. The warheads bind to the E3 ligase and protein of interest (POI), while the flexible linker brings the two proteins into proximity (Qiang et al., 2024).

## A. Related Work

### A.1. Linker Design

#### A.1.1. MOLECULAR LINKER DESIGN

Molecular linker design represents a fundamental challenge in rational drug design, with applications ranging from fragment-based drug discovery (Erlanson et al., 2004) to targeted protein degrader degradation (Chamberlain & Hamann, 2019). Molecular linker design addresses the challenge of connecting molecular fragments through optimal linking groups while preserving or enhancing the desired molecular properties.

Early approaches like SyntaLinker (Yang et al., 2020) treated linker design as a sequence completion problem using SMILES representation (Weininger, 1988). While innovative, this approach suffered from inherent limitations due to its reliance on 1D molecular representations and the absence of crucial 3D structural information.

Subsequent methods addressed these limitations by incorporating spatial information through graph-based approaches. DeLinker (Imrie et al., 2020) marked significant progress by utilizing molecular graphs and incorporating basic geometric constraints such as inter-atomic distances and angles between anchor atoms. However, these methods still operated primarily in 2D space, limiting their ability to capture the full complexity of molecular interactions.

The field has recently witnessed a paradigm shift toward direct 3D linker generation. Methods like 3DLinker (Huang et al., 2022) and DiffLinker (Igashov et al., 2024) leverage advanced deep learning architectures - conditional VAE and diffusion models, respectively – to generate linkers directly in 3D space. LinkerNet (Guan et al., 2023) further operates under the assumption of known fragment poses.

#### A.1.2. PROTAC LINKER DESIGN

PROTAC (PROteolysis TArgeting Chimeras) linker design presents distinct challenges that set it apart from traditional molecular linker design. Since the first proof-of-concept study (Sakamoto et al., 2001), PROTAC development has emerged as a promising therapeutic strategy, distinguished by its ability to induce protein degradation rather than mere inhibition.

PROTAC linker design represents a specialized subset of the molecular linker design problem with distinct requirements and additional complexity. As shown in Figure 6, a PROTAC molecule consists of three key components: a target protein ligand, an E3 ligase ligand, and a linking structure. The design task requires generating a linker that connects these two ligands while enabling the formation of a functional ternary complex with both target proteins.

Traditional PROTAC linker design has largely relied on empirical optimization strategies, typically utilizing a limited repertoire of chemical motifs (Troup et al., 2020). This empirical approach, while pragmatic, fails to fully exploit the

potential chemical space and may miss optimal linker configurations.

Recent attempts to rationalize PROTAC design through computational methods, such as deep reinforcement learning approaches (Zheng et al., 2022; Neeser et al., 2023), represent important steps forward. However, these methods still primarily operate in SMILES space rather than 3D space, limiting their ability to account for the complex spatial requirements of ternary complex formation.

### A.2. Diffusion Generative Models

Diffusion models have emerged as powerful generative models for molecular generation tasks, particularly in the context of 3D molecular structures (Alakhdar et al., 2024; Song et al., 2025; 2024). Inspired by nonequilibrium statistical physics, these models can generate 3D molecular structures with specific properties or requirements crucial to drug discovery. Diffusion models were particularly successful at learning the complex probability distributions of 3D molecular geometries and their corresponding chemical and physical properties, such as conformation generation (Jing et al., 2022), molecular docking (Corso et al., 2023), antibody design (Luo et al., 2022) and protein-ligand binding affinity prediction (Jin et al., 2023).

The first deployment of diffusion models for 3D molecular generation is the E(3) Equivariant Diffusion Model (EDM) (Hoogeboom et al., 2022), where they use an E(n) equivariant graph neural network (EGNN) (Satorras et al., 2021) developed originally for discriminative tasks to denoise and learn molecular structure distributions. Subsequently, they become widely adapted for 3D molecular generation tasks combined with GNNs or transformer-based models for encoding and learning molecule structures (Zhang et al., 2023).

### A.3. Fine-tuning Diffusion Models

Fine-tuning pre-trained diffusion models on limited data from the target domain presents significant challenges, particularly in maintaining generation diversity while avoiding overfitting. Several approaches have been proposed to address these limitations. A novel time-aware adapter architecture within the attention blocks is developed (Moon et al., 2022), which substantially reduces the number of trainable parameters. TGDP further reduces the sample complexity for fine-tuning the pre-trained diffusion model via an additional guidance network for transfer learning (Ouyang et al., 2024). The work in selective parameter fine-tuning (Xie et al., 2023) also demonstrates success by restricting updates to specific model components, including bias terms, class embeddings, normalization layers, and scale factors.

While existing fine-tuning methods for diffusion models have proven effective in image generation (Han et al., 2023; Xie et al., 2023; Ruiz et al., 2023; Zheng et al., 2024; Luo et al., 2023; Zhang et al., 2022; Denker et al., 2024; Uehara et al., 2024; Zeng et al., 2024), they face fundamental limitations in molecular linker design. Image-based approaches often employ techniques such as pairwise similarity loss and high-frequency component regularization to enhance diversity. However, unlike image generation—where minor visual artifacts may be tolerable—molecular structures require strict chemical validity. Regularization strategies designed for image diversity do not inherently preserve valid chemical structures. Moreover, molecular conformations, particularly in PROTACs, involve complex geometric constraints in chemical space that differ fundamentally from feature spaces in image-based models. MoleculeJAE (Du et al., 2023) addresses general molecule generation through self-supervised learning during pre-training, capturing inherent chemical structures. However, the intrinsic differences between small molecules and PROTACs still pose challenges for its adaptation to PROTAC linker design.

## B. Preliminaries and Details of the Proposed DAD-PROTAC Model

In this section, we present some preliminaries and details of our proposed model DAD-PROTAC.

### B.1. Small molecules and PROTACs

#### B.1.1. Difference in Chemical Space

PROTACs, with their bifunctional nature and larger size, present distinct challenges in linker design compared to traditional small molecules. Their higher molecular weights and increased conformational flexibility are critical for effective function. Notably, the physicochemical properties and pharmacokinetic (PK) profiles of PROTACs differ substantially from those of conventional small molecules. Key differences in chemical space include molecular weight (MW), partition coefficient

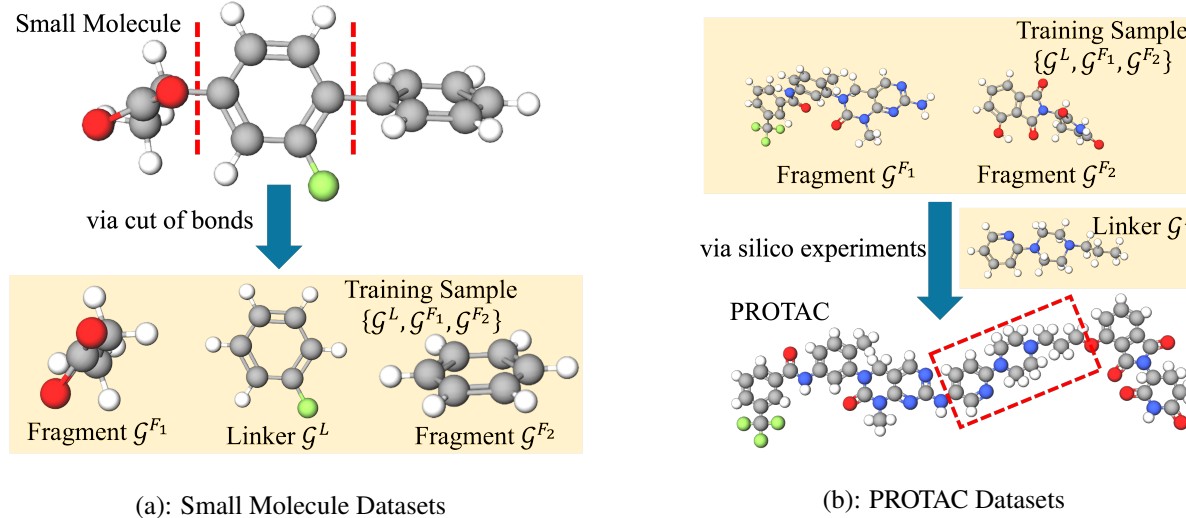

(a): Small Molecule Datasets

(b): PROTAC Datasets

Figure 7: Differences between how small molecule datasets and PROTAC datasets are collected.

(LogP), rotatable bond count (flexibility and conformational dynamics), hydrogen bond donors (HBDs) and acceptors (HBAs), and carbon atom count.

As shown in Figure 2, small molecules typically fall within the 250–500 Da range, aligning with Lipinski's Rule of Five (Chen et al., 2020), which favors oral bioavailability. In contrast, PROTACs peak at 750–1000 Da, emphasizing their significantly larger size. The small-molecule distribution is narrow and sharply peaked, indicating a relatively uniform MW range, whereas the broader PROTAC distribution reflects greater variability in molecular size. This distinct separation underscores a fundamental difference between PROTACs and traditional small molecules: their larger and more diverse structural profiles.

We include more results and analysis on key molecular descriptors in Appendix F.1 to comprehensively characterize the distinct properties of PROTACs compared to traditional small molecules. These molecular descriptor analyses highlight the fundamental differences between PROTACs and traditional small molecules in the chemical space, providing guidelines for PROTAC linker design.

### B.1.2. DIFFERENCE IN DATA COLLECTION

The construction of training datasets for small molecule linkers and PROTAC linkers differs significantly, as illustrated in Figure 7. For small molecules, linker extraction typically follows a systematic fragmentation approach, where acyclic single bonds outside functional groups are algorithmically identified and double-cut to generate fragment-linker pairs. This automated strategy produces large, diverse datasets that comprehensively represent the chemical space of drug-like molecules.

In contrast, PROTAC linker datasets are curated through more specialized and deliberate methods. Rather than retrospective fragmentation, PROTAC linkers are primarily sourced from two main channels: in silico design experiments and manual extraction from published literature. This distinction reflects the rational, design-driven nature of PROTAC development. Unlike small molecule linkers, which benefit from large-scale historical databases, PROTAC linker datasets are inherently smaller and represent engineered solutions in the real world. These fundamental differences in data availability and collection methods must be carefully considered when developing domain adaptation models for PROTAC linker design.

### B.2. E(n) Equivariant Graph Neural Networks

E(n) Equivariant Graph Neural Network (EGNN) (Satorras et al., 2021) is a special type of graph neural network (Song et al., 2023b; Ma et al., 2023; Liang et al., 2022; Song et al., 2023a; 2022) that satisfies the equivariant constraints specified as follows. We consider point clouds $X = (x_1, \cdots, x_N) \in \mathbb{R}^{N \times 3}$ with corresponding features $H = (h_1, \cdots, h_N) \in \mathbb{R}^{N \times nf}$, where $nf$ is the dimension of the node feature. The features $H$ are invariant to group transformations, while the positions $X$

are equivariant to rotations, reflections, and translations. We define the function $(Z_x, Z_h) = f(X, H)$ to be equivariant if for all $R$ and $t$, we have

$$RZ_x + t, Z_h = f(RX + t, H).$$

Considering the interactions among all atoms, we assume a fully connected graph $\mathcal{G}$ with nodes $v_i \in \mathcal{V}$. Each node $v_i$ is associated with coordinates $x_i \in \mathbb{R}^3$ and features $h_i \mathcal{R}^d$. EGNN consists of a composition of multiple Equivariant Graph Convolutional Layers (EGCL), i.e., $X^{l+1}, H^{l+1} = \text{EGCL}[X^l, H^l]$ which are defined as follows.

$$m_{ij} = \phi_e \left( h_j^l, h_i^l, d_{ij}^2, a_{ij} \right).$$

$$h_i^{l+1} = \phi_h \left( h_i^l, \sum_{j \neq i} \tilde{e}_{ij} m_{ij} \right)$$

$$x_i^{l+1} = x_i^l + \sum_{j \neq i} \frac{x_i^l - x_j^l}{d_{ij} + 1} \phi_x \left( h_i^l, h_j^l, d_{ij}^2, a_{ij} \right)$$

Here, $l$ indexes the layers of EGCL. $d_{ij} = \|x_i^l - x_j^l\|$ is the Euclidean distance between nodes $v_i$ and $v_j$. $a_{ij}$ denotes optional edge attributes. The attention mechanism is used to infer a soft estimation of the edges $\tilde{e}_{ij} = \phi_{\text{inf}}(m_{ij})$. All these learnable functions are parameterized by fully connected neural networks. Therefore, with this definition, the entire architecture of EGNN is composed of $L$ EGCN layers as $\hat{X}, \hat{H} = \text{EGNN}\left[X^0, H^0\right]$.

### B.3. Concrete Score

The concrete score is a generalization of the (Stein) score for discrete settings (Meng et al., 2024). Given a predefined neighborhood structure, the Concrete score of any input is defined by the rate of change of the probabilities with respect to local directional changes of the input. This formulation allows us to recover the (Stein) score in continuous domains when measuring such changes by the Euclidean distance.

To be more precise, let $p(x)$ be the data distribution over $\mathcal{X}$. We denote $\mathcal{N} : \mathcal{X} \to \mathcal{X}^K$ as the function mapping each sample $x \in \mathcal{X}$ to a set of neighbors, such that $\mathcal{N}(x) = \{x_{n_1}, \cdots, x_{n_k}\}$ and $K = |\mathcal{N}((x)|$. This neighborhood induces a particular graphical structure onto the support of $p(x)$, which is called the neighborhood-induced graph $\mathcal{G}$. Formally, $\mathcal{G}$ results from adding a directed edge from x to each node in its neighborhood set $x_n \in \mathcal{N}(x)$ for all $x \in \text{supp}(p(x))$.

The Concrete score is constructed as the rate of change of the probabilities with respect to these local directional changes in the input x as defined below.

**Definition B.1.** (Concrete score) Let $\mathcal{N}$ be a function mapping each data point $x$ to its set of neighbors $\mathcal{N}(x) = \{x_{n_1}, \cdots, x_{n_k}\}$. Then the concrete score $c_{p(x)}(x) : \mathcal{X} \to \mathbb{R}^{|\mathcal{N}(x)|}$ for a given distribution $p(x)$ evaluated at $x$ is given as

$$c_{p(x)}(x) := \left[ \frac{p(x_{n_1}) - p(x)}{p(x)}, \cdots, \frac{p(x_{n_k}) - p(x)}{p(x)} \right]^\mathsf{T}.$$

Similar to denoising score matching for score estimation (Vincent, 2011), a denoising counterpart for concrete score estimation is named denoising concrete score matching (Meng et al., 2024). Given a discrete data distribution $p(x)$ and a discrete noise distribution $\tilde{q}(\tilde{x} \mid x)$, the perturbed data distribution $\tilde{p}(\tilde{x}) = \sum_x p(x)\tilde{q}(\tilde{x} \mid x)$, and the posterior $q(x \mid \tilde{x}) = \frac{p(x)\tilde{q}(\tilde{x}|x)}{\tilde{p}(\tilde{x})}$. Then, the concrete score of the perturbed data distribution $\tilde{p}(\tilde{x})$ can be obtained via $x_{\tilde{p}(\tilde{x})}(\tilde{x}) = \sum_x c_{\tilde{q}(\tilde{x}|x)}(\tilde{x})q(x \mid \tilde{x})$. This property leads to the following objective.

**Theorem B.2.** *If the following objective is minimized,*

$$\mathcal{L}^{D-CSM}(\theta) = \sum_{x, \tilde{x}} p(x)\tilde{q}(\tilde{x} \mid x)\|c_\theta(\tilde{x}) - c_{q(\tilde{x}|x)}(\tilde{x})\|_2^2,$$

*we have* $c_\theta(\tilde{x}) = c_{p(\tilde{x})}(\tilde{x})$.

**B.4. Learning the Concrete Score Terms**

Recall that each PROTAC linker can be represented as a triple $\mathcal{G}^L = \{X^L, A^L, B^L\}$. In Section 2.3.2, we have discussed how to learn the concrete score correction term for $X^L$ for the target PROTAC domain during the fine-tuning phase. In this section, we will discuss how to learn the concrete score correction terms for $A^L$ and $B^L$ in the same manner.

We show that the concrete score estimators $c_{\tilde{\phi}^*}(A_t^L, t), c_{\tilde{\phi}^*}(B_t^L, t)$ for the target PROTAC domain during the fine-tuning phase can be decomposed as

$$c_{\tilde{\phi}^*}(A_t^L, t) = c_{q(A_t^L)}(A_t^L) + c_{q(A_t^L)}\left(\mathbb{E}_{q(A_0^L|A_t^L)}\frac{\tilde{q}(A_0^L)}{q(A_0^L)}\right). \tag{17}$$

$$c_{\tilde{\phi}^*}(B_t^L, t) = c_{q(B_t^L)}(B_t^L) + c_{q(B_t^L)}\left(\mathbb{E}_{q(B_0^L|B_t^L)}\frac{\tilde{q}(B_0^L)}{q(B_0^L)}\right). \tag{18}$$

Similarly, we still focus on estimating the density ratio term in the chemical space $\frac{\tilde{q}(A_0^L)}{q(A_0^L)}$ ($\frac{\tilde{q}(B_0^L)}{q(B_0^L)}$) since it is involved in the concrete score correction term in Equation (17) (Equation (18)).

Based on Theorem 2.2, we can figure out how the neural network $\psi^*(A_t^L, t)$ should be trained to approximate the density ratio term $\mathbb{E}_{q(A_0^L|A_t^L)}\frac{\tilde{q}(A_0^L)}{q(A_0^L)}$. Namely, we have

$$\psi^*(A_t^L, t) = \arg\min \mathcal{L}^{\psi_A} = \mathbb{E}_{q(A_0^L|A_t^L)}\frac{\tilde{q}(A_0^L)}{q(A_0^L)},$$

where the loss is set as

$$\mathcal{L}^{\psi_A} = \mathbb{E}_{\tilde{q}(A_0^L, A_t^L)}\left\|\psi(A_t^L, t) - \frac{\tilde{q}(A_t^L)}{q(A_t^L)}\right\|_2^2. \tag{19}$$

And we have,

$$\psi^*(B_t^L, t) = \arg\min \mathcal{L}^{\psi_B} = \mathbb{E}_{q(B_0^L|B_t^L)}\frac{\tilde{q}(B_0^L)}{q(B_0^L)},$$

where the loss is set as

$$\mathcal{L}^{\psi_B} = \mathbb{E}_{\tilde{q}(B_0^L, B_t^L)}\left\|\psi(B_t^L, t) - \frac{\tilde{q}(B_t^L)}{q(B_t^L)}\right\|_2^2. \tag{20}$$

Therefore, the final score estimators for the target PROTAC domain can be calculated as follows.

$$s_{\tilde{\phi}^*}(A_t^L, t) = c_{q(A_t^L)}(A_t^L) + c_{q(A_t^L)}\left(\psi^*(A_t^L, t)\right). \tag{21}$$

$$s_{\tilde{\phi}^*}(B_t^L, t) = c_{q(B_t^L)}(B_t^L) + c_{q(B_t^L)}\left(\psi^*(B_t^L, t)\right). \tag{22}$$

Additionally, in order to estimate the density ratio term $\frac{\tilde{q}(A_t^L)}{q(A_t^L)}$ in Equation (19) and $\frac{\tilde{q}(B_t^L)}{q(B_t^L)}$ in Equation (20), we still need to train another two binary classifiers $\mathcal{W}(A_t^L, t)$ via the cross-entropy loss in Equation (23) and $\mathcal{W}(B_t^L, t)$ via the cross-entropy loss in Equation (24) to determine whether the perturbed samples are from the source small molecule domain or the target PROTAC domain.

$$\mathcal{L}^{\mathcal{W}_A} = -\frac{1}{m}\sum_{A_0^L \sim q(A_0^L)}\sum_{A_t^L \sim q(A_t^L|A_0^L)}\log\mathcal{W}(A_t^L, t) - \frac{1}{n}\sum_{A_0^L \sim \tilde{q}(A_0^L)}\sum_{A_t^L \sim \tilde{q}(A_t^L|A_0^L)}\log\left(1 - \mathcal{W}(A_t^L, t)\right) \tag{23}$$

$$\mathcal{L}^{\mathcal{W}_B} = -\frac{1}{m}\sum_{B_0^L \sim q(B_0^L)}\sum_{B_t^L \sim q(B_t^L|B_0^L)}\log\mathcal{W}(B_t^L, t) - \frac{1}{n}\sum_{B_0^L \sim \tilde{q}(B_0^L)}\sum_{B_t^L \sim \tilde{q}(B_t^L|B_0^L)}\log\left(1 - \mathcal{W}(B_t^L, t)\right) \tag{24}$$

The density ratio is approximated as

$$\frac{\tilde{q}(A_t^L)}{q(A_t^L)} = \frac{1 - \mathcal{W}(A_t^L, t)}{\mathcal{W}(A_t^L, t)}, \quad \frac{\tilde{q}(B_t^L)}{q(B_t^L)} = \frac{1 - \mathcal{W}(B_t^L, t)}{\mathcal{W}(B_t^L, t)}.$$

## C. Proofs

### C.1. Theorem 2.1

*Proof.* We focus on proving the decomposition of the score estimator $s_{\tilde{\phi}^*}(X_t^L, t)$ and the proof for the case of the concrete score estimator $c_{\tilde{\phi}^*}(A_t^L, t)$ follows the exact same manner.

Since we assume that $q(\mathcal{G}_t^L \mid \mathcal{G}_0^L) = \tilde{q}(\mathcal{G}_t^L \mid \mathcal{G}_0^L)$, we know

$$q(X_t^L | X_0^L) = \tilde{q}(X_t^L | X_0^L).$$

Similar to the training loss for the pre-training phase in Equation (5) on the source small molecule domain, the training loss for directly fine-tuning on the target PROTAC domain should be

$$\tilde{\mathcal{L}}^L = \mathbb{E}_{t \sim \mathcal{U}(1,T)} \mathbb{E}_{X_0^L \sim \tilde{q}(X_0^L)} \mathbb{E}_{X_t^L \sim \tilde{q}(X_t^L | X_0^L)} \| s_{\tilde{\phi}}(X_t^L, t) - \nabla_{X_t^L} \log \tilde{q}(X_t^L \mid X_0^L) \|_2^2.$$

We then have

$$\begin{aligned}
s_{\tilde{\phi}^*}(X_t^L, t) &= \arg\min \mathbb{E}_{t \sim \mathcal{U}(1,T)} \mathbb{E}_{X_0^L \sim \tilde{q}(X_0^L)} \mathbb{E}_{X_t^L \sim \tilde{q}(X_t^L | X_0^L)} \| s_{\tilde{\phi}}(X_t^L, t) - \nabla_{X_t^L} \log \tilde{q}(X_t^L \mid X_0^L) \|_2^2 \\
&= \arg\min \mathbb{E}_t \left\{ \mathbb{E}_{\tilde{q}(X_0^L)} \mathbb{E}_{\tilde{q}(X_t^L | X_0^L)} \left[ \left\| s_{\tilde{\phi}}(X_t^L, t) - \nabla_{X_t^L} \log \tilde{q}(X_t^L \mid X_0^L) \right\|_2^2 \right] \right\}
\end{aligned}$$

Then, we only need to focus on $\phi$ related terms

$$\begin{aligned}
&\mathbb{E}_{\tilde{q}(X_0^L)} \mathbb{E}_{\tilde{q}(X_t^L | X_0^L)} \left[ \left\| s_{\tilde{\phi}}(X_t^L, t) - \nabla_{X_t^L} \log \tilde{q}(X_t^L \mid X_0^L) \right\|_2^2 \right] \\
&= \mathbb{E}_{\tilde{q}(X_0^L, X_t^L)} \left[ \left\| s_{\tilde{\phi}}(X_t^L, t) \right\|_2^2 \right] - 2 \mathbb{E}_{\tilde{q}(X_0^L, X_t^L)} \left[ \langle s_{\tilde{\phi}}(X_t^L, t), \nabla_{X_t^L} \log \tilde{q}(X_t^L \mid X_0^L) \rangle \right] + C_1,
\end{aligned}$$

The first term is

$$\begin{aligned}
\mathbb{E}_{\tilde{q}(X_0^L, X_t^L)} \left[ \left\| s_{\tilde{\phi}}(X_t^L, t) \right\|_2^2 \right] &= \int_{X_0^L} \int_{X_t^L} \tilde{q}(X_0^L, X_t^L) \left\| s_{\tilde{\phi}}(X_t^L, t) \right\|_2^2 dX_0^L dX_t^L \\
&= \int_{X_0^L} \int_{X_t^L} q(X_0^L) \tilde{q}(X_t^L | X_0^L) \left\| s_{\tilde{\phi}}(X_t^L, t) \right\|_2^2 \frac{\tilde{q}(X_0^L)}{q(X_0^L)} dX_0^L dX_t^L \\
&= \int_{X_0^L} \int_{X_t^L} q(X_0^L) q(X_t^L | X_0^L) \left\| s_{\tilde{\phi}}(X_t^L, t) \right\|_2^2 \frac{\tilde{q}(X_0^L)}{q(X_0^L)} dX_0^L dX_t^L \\
&= \mathbb{E}_{q(X_0^L, X_t^L)} \left[ \left\| s_{\tilde{\phi}}(X_t^L, t) \right\|_2^2 \frac{\tilde{q}(X_0^L)}{q(X_0^L)} \right]
\end{aligned}$$

The second term is

$$
\mathbb{E}_{\tilde{q}(X_0^L, X_t^L)} \left[ \langle s_{\tilde{\phi}}(X_t^L, t), \nabla_{X_t^L} \log \tilde{q}(X_t^L \mid X_0^L) \rangle \right]
$$

$$
= \int_{X_0^L} \int_{X_t^L} \langle s_{\tilde{\phi}}(X_t^L, t), \nabla_{X_t^L} \log \tilde{q}(X_t^L | X_0^L) \rangle \tilde{q}(X_t^L | X_0^L) \tilde{q}(X_0^L) dX_0^L dX_t^L
$$

$$
= \int_{X_0^L} \int_{X_t^L} \langle s_{\tilde{\phi}}(X_t^L, t), \nabla_{X_t^L} \tilde{q}(X_t^L | X_0^L) \rangle \tilde{q}(X_0^L) dX_0^L dX_t^L
$$

$$
= \int_{X_0^L} \int_{X_t^L} q(X_0^L) q(X_t^L | X_0^L) \langle s_{\tilde{\phi}}(X_t^L, t), \frac{\nabla_{X_t^L} \tilde{q}(X_t^L | X_0^L)}{q(X_t^L | X_0^L)} \rangle \frac{\tilde{q}(X_0^L)}{q(X_0^L)} dX_0^L dX_t^L
$$

$$
= \int_{X_0^L} \int_{X_t^L} q(X_0^L, X_t^L) \langle s_{\tilde{\phi}}(X_t^L, t), \frac{\nabla_{X_t^L} q(X_t^L | X_0^L)}{q(X_t^L | X_0^L)} \rangle \frac{\tilde{q}(X_0^L)}{q(X_0^L)} dX_0^L dX_t^L
$$

$$
= \mathbb{E}_{q(X_0^L, X_t^L)} \left[ \langle s_{\tilde{\phi}}(X_t^L, t), \nabla_{X_t^L} \log q(X_t^L | X_0^L) \rangle \frac{\tilde{q}(X_0^L)}{q(X_0^L)} \right]
$$

Therefore, we have

$$
\arg\min \mathbb{E}_{\tilde{q}(X_0^L)} \mathbb{E}_{\tilde{q}(X_t^L | X_0^L)} \left[ \left\| s_{\tilde{\phi}}(X_t^L, t) - \nabla_{X_t^L} \log \tilde{q}(X_t^L \mid X_0^L) \right\|_2^2 \right]
$$

$$
= \arg\min \mathbb{E}_{\tilde{q}(X_0^L, X_t^L)} \left[ \left\| s_{\tilde{\phi}}(X_t^L, t) \right\|_2^2 \right] - 2\mathbb{E}_{\tilde{q}(X_0^L, X_t^L)} \left[ \langle s_{\tilde{\phi}}(X_t^L, t), \nabla_{X_t^L} \log \tilde{q}(X_t^L \mid X_0^L) \rangle \right] + C_1
$$

$$
= \arg\min \mathbb{E}_{q(X_0^L, X_t^L)} \left[ \left\| s_{\tilde{\phi}}(X_t^L, t) \right\|_2^2 \frac{\tilde{q}(X_0^L)}{q(X_0^L)} \right] - 2\mathbb{E}_{q(X_0^L, X_t^L)} \left[ \langle s_{\tilde{\phi}}(X_t^L, t), \nabla_{X_t^L} \log q(X_t^L | X_0^L) \rangle \frac{\tilde{q}(X_0^L)}{q(X_0^L)} \right] + C_2
$$

$$
= \arg\min \mathbb{E}_{q(X_0^L)} \mathbb{E}_{q(X_t^L | X_0^L)} \left[ \left\| s_{\tilde{\phi}}(X_t^L, t) - \nabla_{X_t^L} \log q(X_t^L | X_0^L) \right\|_2^2 \frac{\tilde{q}(X_0^L)}{q(X_0^L)} \right]
$$

Therefore, we have,

$$
s_{\tilde{\phi}^*}(X_t^L, t) = \arg\min \mathbb{E}_t \left\{ \mathbb{E}_{q(X_0^L)} \mathbb{E}_{q(X_t^L | X_0^L)} \left[ \left\| s_{\tilde{\phi}}(X_t^L, t) - \nabla_{X_t^L} \log q(X_t^L | X_0^L) \right\|_2^2 \frac{\tilde{q}(X_0^L)}{q(X_0^L)} \right] \right\}.
$$

Then, we can use Importance Weighted Denoising Score Matching on the source small molecule domain and thus get the closed-form of $s_{\tilde{\phi}^*}(X_t^L, t)$ as follows:

$$
s_{\tilde{\phi}^*}(X_t^L, t) = \frac{\mathbb{E}_{q(X_0^L | X_t^L)} \left[ \nabla_{X_t^L} \log q(X_t^L | X_0^L) \frac{q(X_0^L)}{q(X_0^L)} \right]}{\mathbb{E}_{q(X_0^L | X_t^L)} \left[ \frac{q(X_0^L)}{q(X_0^L)} \right]}.
$$

Note that we also have

$$
\nabla_{X_t^L} \log q(X_0^L | X_t^L) = \nabla_{X_t^L} \log q(X_t^L | X_0^L) + \nabla_{X_t^L} \log q(X_0^L) - \nabla_{X_t^L} \log q(X_t^L)
$$

$$
= \nabla_{X_t^L} \log q(X_t^L | X_0^L) - \nabla_{X_t^L} \log q(X_t^L),
$$

Finally, we have that

$$\mathbf{s}_{\phi^*}(X_t^L, t)$$

$$=\frac{\mathbb{E}_{q(X_0^L|X_t^L)}\left[\nabla_{X_t^L}\log q(X_t^L|X_0^L)\frac{\tilde{q}(X_0^L)}{q(X_0^L)}\right]}{\mathbb{E}_{q(X_0^L|X_t^L)}\left[\frac{\tilde{q}(X_0^L)}{q(X_0^L)}\right]}$$

$$=\nabla_{X_t^L}\log q(X_t^L) + \frac{\mathbb{E}_{q(X_0^L|X_t^L)}\left[\frac{\tilde{q}(X_0^L)}{q(X_0^L)}\nabla_{X_t^L}\log q(X_t^L|X_0^L)\right]}{\mathbb{E}_{q(X_0^L|X_t^L)}\left[\frac{\tilde{q}(X_0^L)}{q(X_0^L)}\right]} - \nabla_{X_t^L}\log q(X_t^L)$$

$$=\nabla_{X_t^L}\log q(X_t^L) + \frac{\mathbb{E}_{q(X_0^L|X_t^L)}\left[\frac{\tilde{q}(X_0^L)}{q(X_0^L)}\left(\nabla_{X_t^L}\log q(X_0^L|X_t^L) + \nabla_{X_t^L}\log q(X_t^L)\right)\right]}{\mathbb{E}_{q(X_0^L|X_t^L)}\left[\frac{\tilde{q}(X_0^L)}{q(X_0^L)}\right]} - \nabla_{X_t^L}\log q(X_t^L)$$

$$=\nabla_{X_t^L}\log q(X_t^L) + \frac{\mathbb{E}_{q(X_0^L|X_t^L)}\left[\frac{\tilde{q}(X_0^L)}{q(X_0^L)}\nabla_{X_t^L}\log q(X_0^L|X_t^L)\right]}{\mathbb{E}_{q(X_0^L|X_t^L)}\left[\frac{\tilde{q}(X_0^L)}{q(X_0^L)}\right]}$$

$$=\nabla_{X_t^L}\log q(X_t^L) + \frac{\nabla_{X_t^L}\mathbb{E}_{q(X_0^L|X_t^L)}\left[\frac{\tilde{q}(X_0^L)}{q(X_0^L)}\right]}{\mathbb{E}_{q(X_0^L|X_t^L)}\left[\frac{\tilde{q}(X_0^L)}{q(X_0^L)}\right]}$$

$$=\nabla_{X_t^L}\log q(X_t^L) + \nabla_{X_t^L}\log \mathbb{E}_{q(X_0^L|X_t^L)}\left[\frac{\tilde{q}(X_0^L)}{q(X_0^L)}\right]$$

To sum up, we prove that

$$s_{\tilde{\phi}^*}(X_t^L, t) = \nabla_{X_t^L}\log q(X_t^L) + \nabla_{X_t^L}\log \mathbb{E}_{q(X_0^L|X_t^L)}\frac{\tilde{q}(X_0^L)}{q(X_0^L)}.$$

Similarly, for the case of the concrete score estimator $c_{\tilde{\phi}^*}(A_t^L, t)$, we can also prove that

$$c_{\tilde{\phi}^*}(A_t^L, t) = c_{q(A_t^L)}(A_t^L) + c_{q(A_t^L)}\left(\mathbb{E}_{q(A_0^L|A_t^L)}\frac{\tilde{q}(A_0^L)}{q(A_0^L)}\right).$$

$\square$

### C.2. Theorem 2.2

*Proof.* We first note that the objective function $\mathcal{L}^{\psi_X}$ can be rewritten as follows.

$$\psi^*(X_t^L, t) = \arg\min \mathcal{L}^{\psi_X}$$

$$= \arg\min \mathbb{E}_{\tilde{q}(X_0^L, X_t^L)}\left\|\psi(X_t^L, t) - \frac{\tilde{q}(X_t^L)}{q(X_t^L)}\right\|_2^2$$

$$= \arg\min \int_{X_t^L}\left\{\int_{X_0^L}\tilde{q}\left(X_0^L \mid X_t^L\right)\left\|\psi\left(X_t^L, t\right) - \frac{\tilde{q}\left(X_t^L\right)}{q\left(X_t^L\right)}\right\|_2^2 dX_0^L\right\}\tilde{q}\left(X_t^L\right)dX_t^L$$

$$= \arg\min \int_{X_t^L}\left\{\left\|\psi\left(X_t^L, t\right)\right\|_2^2 - 2\left\langle\psi\left(X_t^L, t\right), \int_{X_0^L}\tilde{q}\left(X_0^L \mid X_t^L\right)\frac{\tilde{q}\left(X_t^L\right)}{q\left(X_t^L\right)}dX_0^L\right\rangle\right\}\tilde{q}\left(X_t^L\right)dX_t^L + C$$

$$= \arg\min \int_{X_t^L}\left\|\psi\left(X_t^L, t\right) - \mathbb{E}_{\tilde{q}\left(X_0^L|X_t^L\right)}\left[\frac{\tilde{q}\left(X_t^L\right)}{q\left(X_t^L\right)}\right]\right\|_2^2\tilde{q}\left(X_t^L\right)dX_t^L.$$

Here, $C$ is a constant independent of $\psi$. Hence, we have the minimize $\psi^*(X_t^L, t) = \arg\min \mathcal{L}^{\psi_X}$ satisfies the following.

$$\psi^*(X_t^L, t) = \arg\min \mathcal{L}^{\psi_X} = \mathbb{E}_{\tilde{q}(X_0^L | X_t^L)} \left[ \frac{\tilde{q}(X_t^L)}{q(X_t^L)} \right] \tag{25}$$

Also, we can rewrite the term in the RHS of Equation (14) as follows.

$$
\begin{aligned}
\mathbb{E}_{q(X_0^L | X_t^L)} \left[ \frac{\tilde{q}(X_0^L)}{q(X_0^L)} \right] &= \int q(X_0^L \mid X_t^L) \frac{\tilde{q}(X_0^L)}{q(X_0^L)} dX_0^L \\
&= \int \frac{q(X_t^L \mid X_0^L) q(X_0^L)}{q(X_t^L)} \frac{\tilde{q}(X_0^L)}{q(X_0^L)} dX_0^L \\
&= \int \frac{\tilde{q}(X_t^L \mid X_0^L) q(X_0^L)}{q(X_t^L)} \frac{\tilde{q}(X_0^L)}{q(X_0^L)} dX_0^L \\
&= \int \tilde{q}(X_t^L \mid X_0^L) \frac{\tilde{q}(X_0^L)}{q(X_t^L)} dX_0^L \\
&= \int \frac{\tilde{q}(X_0^L \mid X_t^L) \tilde{q}(X_t^L)}{\tilde{q}(X_0^L)} \frac{\tilde{q}(X_0^L)}{q(X_t^L)} dX_0^L \\
&= \int \tilde{q}(X_0^L \mid X_t^L) \frac{\tilde{q}(X_t^L)}{q(X_t^L)} dX_0^L \\
&= \mathbb{E}_{\tilde{q}(X_0^L | X_t^L)} \left[ \frac{\tilde{q}(X_t^L)}{q(X_t^L)} \right]
\end{aligned}
\tag{26}
$$

Based on Equation (26) and Equation (25), we now prove

$$\psi^*(X_t^L, t) = \arg\min \mathcal{L}^{\psi_X} = \mathbb{E}_{q(X_0^L | X_t^L)} \frac{\tilde{q}(X_0^L)}{q(X_0^L)}$$

$\square$

*Remark* C.1. Similarly, for the case of the $\mathcal{L}^{\psi_A}$ and $\mathcal{L}^{\psi_B}$, we can also prove that

$$\psi^*(A_t^L, t) = \arg\min \mathcal{L}^{\psi_A} = \mathbb{E}_{q(A_0^L | A_t^L)} \frac{\tilde{q}(A_0^L)}{q(A_0^L)},$$

where the loss is set as

$$\mathcal{L}^{\psi_A} = \mathbb{E}_{\tilde{q}(A_0^L, A_t^L)} \left\| \psi(A_t^L, t) - \frac{\tilde{q}(A_t^L)}{q(A_t^L)} \right\|_2^2.$$

And we also have,

$$\psi^*(B_t^L, t) = \arg\min \mathcal{L}^{\psi_B} = \mathbb{E}_{q(B_0^L | B_t^L)} \frac{\tilde{q}(B_0^L)}{q(B_0^L)},$$

where the loss is set as

$$\mathcal{L}^{\psi_B} = \mathbb{E}_{\tilde{q}(B_0^L, B_t^L)} \left\| \psi(B_t^L, t) - \frac{\tilde{q}(B_t^L)}{q(B_t^L)} \right\|_2^2.$$

## D. Pseudocodes

### D.1. Pre-training Phase of the Proposed DAD-PROTAC Model

We summarize the pre-training phase of the proposed DAD-PROTAC model in Algorithm Algorithm 2. The pre-training phase operates on the small molecule domain $q(\mathcal{G})$. The process begins by sampling small molecule data $(A_0^L, B_0^L, X_0^L)$ from these distributions (line 2). For each iteration, we sample a time step t uniformly from $[1, T]$ and apply diffusion noise

---

**Algorithm 2** Pre-training Phase of DAD-PROTAC

---

**input** Samples from the small molecule domain $q(\mathcal{G}) = \big(q(A^L), q(B^L), q(X^L)\big)$, pre-trained diffusion process $q(\mathcal{G}_t^L \mid \mathcal{G}_0^L))$

**output** Score functions $c_{q(A_t^L)}(A_t^L)$, $c_{q(B_t^L)}(B_t^L)$, and $\nabla_{X_t^L} \log q(X_t^L)$

1: **repeat**
2:     Sample the small molecule data from $(A_0^L, B_0^L, X_0^L) \sim \big(q(A^L), q(B^L), q(X^L)\big)$.
3:     Sample the time step from $t \sim \mathcal{U}(1, T)$.
4:     Perturb the previous sampled data $(A_0^L, B_0^L, X_0^L)$ via the corresponding diffusion process $q(\mathcal{G}_t^L \mid \mathcal{G}_0^L)$ based on $q(A_t^L \mid A_0^L)$ in Equation (7), $q(B_t^L \mid B_0^L)$, and $q(X_t^L \mid X_0^L)$ in Equation (2).
5:     Perform gradient descent step over the EGNN model $\phi_\theta(\mathcal{G}^{P_t}, t)$ to predict $c_\theta(A_t^L, t)$ via the loss $\mathcal{L}^A$ in Equation (9), $c_\theta(B_t^L, t)$ via the loss $\mathcal{L}^B$, and $s_\theta(X_t^L, t)$ via the loss $\mathcal{L}^X$ in Equation (5).
6: **until** converged.
7: **return** Score functions $c_{q(A_t^L)}(A_t^L) \approx c_\theta(A_t^L, t)$, $c_{q(B_t^L)}(B_t^L) \approx c_\theta(B_t^L, t)$, and $\nabla_{X_t^L} \log q(X_t^L) \approx s_\theta(X_t^L, t)$ on the small molecule domain for the subsequent fine-tuning phase

---

to the sampled molecules using the corresponding diffusion process $q(\mathcal{G}_t \mid \mathcal{G}_0)$ (line 4). An EGNN model $\phi_\theta$ is then trained through gradient descent to predict three score functions: $c_\theta(A_t^L, t)$, $c_\theta(B_t^L, t)$ and $s_\theta(X_t^L, t)$ (line 5). Upon convergence, the algorithm outputs the learned score functions for each component, which serve as the foundation for the subsequent fine-tuning phase (line 7). In summary, this pre-training phase ensures that the model learns the representations of general linker structures before adaptation to the specific PROTAC domain.

### D.2. Fine-tuning Phase of the Proposed DAD-PROTAC Model

We present the detailed fine-tuning phase of the proposed DAD-PROTAC model in Algorithm 3. The fine-tuning process of DAD-PROTAC comprises two primary steps: density ratio estimation and score correction, designed to efficiently transfer knowledge from small molecules to PROTACs.

In the density ratio estimation step, we begin by sampling $m$ instances from the source small molecule domain $q(\mathcal{G})$ and $n$ instances from the target PROTAC domain $\tilde{q}(\mathcal{G})$ (line 2). We then sample a time step $t$ uniformly from interval $[1, T]$ and apply the forward diffusion process $q(\mathcal{G}_t|\mathcal{G}_0) = \tilde{q}(\mathcal{G}_t|\mathcal{G}_0)$ to perturb the sampled data. Note that the forward diffusion processes in both the small molecule domain and the PROTAC domain are the same (line 3). A binary classifier $\mathcal{W}$ is trained through gradient descent to distinguish between the domains using specified loss functions for each component of the linker $\mathcal{G} = \{A^L, B^L, X^L\}$ (line 4). Upon convergence, we compute the density ratio $\frac{\tilde{q}(\mathcal{G}_t)}{q(\mathcal{G}_t)} = \frac{1 - \mathcal{W}(\mathcal{G}_t, t)}{\mathcal{W}(\mathcal{G}_t, t)}$, which quantifies the distribution differences in the chemical space explicitly (line 6).

In the score correction phase, we leverage another neural network $\psi(\mathcal{G}_t, t)$ to approximate the necessary adjustments to the pre-trained score function as the score correction term. We sample data exclusively from the PROTAC domain $\tilde{q}(\mathcal{G})$ and apply noise perturbation through the diffusion process $\tilde{q}(\mathcal{G}_t \mid \mathcal{G}_0)$ (line 8). The network is trained via gradient descent to learn the score corrections (line 9). The final output is a score estimator specifically adapted for the PROTAC domain, incorporating both the pre-trained knowledge and domain-specific adjustments (line 11).

## E. Experimental Setup Details

### E.1. Datasets

E.1.1. ZINC

Following the setting of Delinker (Imrie et al., 2020), we also consider a subset of 250,000 molecules randomly selected from the ZINC database (Gómez-Bombarelli et al., 2018; Irwin & Shoichet, 2005). We first generate 3D conformers using RDKit (Landrum, 2016) and define a reference structure for each molecule by selecting the lowest-energy conformation. Molecules are then fragmented by enumerating all double cuts of acyclic single bonds outside functional groups. The resulting fragments and linkers are filtered based on atom count, synthetic accessibility score, ring aromaticity, and pan-assay interference compounds (PAINS) criteria (Baell & Holloway, 2010). Since each molecule can yield multiple fragment-linker combinations, the final dataset comprises 438,610 samples, which we use for model pre-training. The dataset includes the

---

**Algorithm 3** Fine-tuning Phase of DAD-PROTAC in Detail

---

**input** Samples from the source small molecule domain $q(\mathcal{G}) = \big(q(A^L), q(B^L), q(X^L)\big)$ and target PROTAC domain $\tilde{q}(\mathcal{G}) = \big(\tilde{q}(A^L), \tilde{q}(B^L), \tilde{q}(X^L)\big)$, pre-trained (fine-tuning) diffusion process $q(\mathcal{G}_t \mid \mathcal{G}_0) = \tilde{q}(\mathcal{G}_t \mid \mathcal{G}_0))$

/* Estimate the density ratio in the chemical space via a binary classifier $\mathcal{W}$ first from lines 1-5. */

1: **repeat**
2:    Sample the small molecule data from $(A_0^L, B_0^L, X_0^L) \sim \big(q(A^L), q(B^L), q(X^L)\big)$ with size of $m$, and sample the PROTAC data from $(A_0^L, B_0^L, X_0^L) \sim \big(\tilde{q}(A^L), \tilde{q}(B^L), \tilde{q}(X^L)\big)$ with size of $n$.
3:    Sample $t \sim \mathcal{U}(1, T)$ and perturb previous sampled data $(A_0^L, B_0^L, X_0^L)$ via the corresponding diffusion process $q(\mathcal{G}_t \mid \mathcal{G}_0) = \tilde{q}(\mathcal{G}_t \mid \mathcal{G}_0)$ based on $q(A_t^L \mid A_0^L)$ in Equation (7), $q(B_t^L \mid B_0^L)$, and $q(X_t^L \mid X_0^L)$ in Equation (2).
4:    Perform gradient descent step over the binary classifiers $\mathcal{W}(A_t^L, t)$ via the loss in Equation (23), $\mathcal{W}(B_t^L, t)$ via the loss in Equation (24), and $\mathcal{W}(X_t^L, t)$ via the loss in Equation (16).
5: **until** converged.
6: Set the density ratio in the chemical space term $\frac{\tilde{q}(A_t^L)}{q(A_t^L)} = \frac{1 - \mathcal{W}(A_t^L, t)}{\mathcal{W}(A_t^L, t)}$ in Equation (19), $\frac{\tilde{q}(B_t^L)}{q(B_t^L)} = \frac{1 - \mathcal{W}(B_t^L, t)}{\mathcal{W}(B_t^L, t)}$ in Equation (20), and $\frac{\tilde{q}(X_t^L)}{q(X_t^L)} = \frac{1 - \mathcal{W}(X_t^L, t)}{\mathcal{W}(X_t^L, t)}$ in Equation (13).

/* Learn to approximate the score correction term for the fine-tuning phase from lines 7-10. */

7: **repeat**
8:    Sample the PROTAC data from $(A_0^L, B_0^L, X_0^L) \sim \big(\tilde{q}(A^L), \tilde{q}(B^L), \tilde{q}(X^L)\big)$ and perturb $(A_0^L, B_0^L, X_0^L)$ with noise via diffusion process based on $\tilde{q}(A_t^L \mid A_0^L)$, $\tilde{q}(B_t^L \mid B_0^L)$, and $\tilde{q}(X_t^L \mid X_0^L)$ with $\tilde{q}(\mathcal{G}_t \mid \mathcal{G}_0) = q(\mathcal{G}_t \mid \mathcal{G}_0)$.
9:    Perform gradient descent step over $\psi(A_t, t)$ via the loss in Equation (19), $\psi(B_t, t)$ via the loss in Equation (20) and $\psi(X_t, t)$ via the loss in Equation (13).
10: **until** converged
11: Set score estimator for the target PROTAC domain $s_{\tilde{\phi}^*}(A_t^L, t)$ via Equation (21), $s_{\tilde{\phi}^*}(B_t^L, t)$ via Equation (22), and $s_{\tilde{\phi}^*}(X_t^L, t)$ via Equation (15). // Combine the pre-trained score estimators to build the score estimator for the PROTAC domain.
12: **return** Score estimator for the target PROTAC domain as $\Big(s_{\tilde{\phi}^*}(A_t^L, t), s_{\tilde{\phi}^*}(B_t^L, t)), s_{\tilde{\phi}^*}(X_t^L, t)\Big)$.

---

following atom types: C, O, N, F, S, Cl, Br, and I.

### E.1.2. PROTAC-DB

PROTAC-DB 2.0 was recently published (Weng et al., 2022), which contains basic information on 3,270 PROTACs. The chemical structures, biological activities, physiochemical properties, and pharmacokinetic parameters of these compounds are manually extracted from the literature or calculated by some programs. Following the data collection method in DiffPROTAC (Li et al., 2024), the simplified molecular-input line-entry system representations for the E3 ligands, the warheads (ligands that bind to targets), and the linkers of each PROTAC from the corresponding pages on the PROTAC-DB website are gathered. This process results in a final dataset of 365 warheads, 82 E3 ligands, 1501 linkers, and 3,270 PROTACs. We use the PROTAC-DB dataset for fine-tuning the model.

### E.2. Baselines

#### E.2.1. 3DLINKER

3DLinker (Huang et al., 2022) is a conditional generative model for the linker design task, and is able to predict anchor atoms and jointly generate linker graphs and their 3D structures based on an E(3) equivariant graph variational autoencoder. We train it for 20 epochs on the ZINC dataset. We also apply for a permutation of the given two fragments to enhance the model performance. For the 3Dlinker-Fine-tuning model, we further directly fine-tune it on the PROTAC-DB dataset.

#### E.2.2. DIFFLINKER

DiffLinker (Igashov et al., 2024) is an E(3)-equivariant three-dimensional conditional diffusion model for molecular linker design. It can link an arbitrary number of fragments. We set the dimension of the hidden embedding as 256 with a 5-layer EGNN. Similarly, for the DiffLinker-Fine-tuning model, we further directly fine-tune it on the PROTAC-DB dataset.

### E.2.3. LINKERNET

LinkerNet (Guan et al., 2023) is another 3D equivariant diffusion model that jointly learns the generative process of both fragment poses and the 3D structure of the linker. In each layer, the input features are concatenated and the hidden embedding/positions are updated with a 2-layer MLP with the LayerNorm and the ReLU activation. For the LinkerNet-Fine-tuning model, we further directly fine-tune it on the PROTAC-DB dataset.

## E.3. Metrics

Our evaluation metrics include validity, uniqueness, and recovery. To assess the results, we employ a process where we sample 100 conformations for each input ligand pair in the test set and subsequently calculate the following metrics.

### E.3.1. VALIDITY

This metric evaluates the validity of generated PROTAC molecules, ensuring they remain in the chemical space. We use OpenBabel (O'Boyle et al., 2011) to determine bond connectivity and RDKit (Landrum, 2016) to verify compliance with valency rules. Additionally, validity requires the absence of dissociative atoms and the preservation of specified fragments. Since fragments are fixed as conditions during learning, our assessment focuses on detecting any detached atoms.

### E.3.2. UNIQUENESS

To quantitatively assess the diversity of generated molecules, we define a uniqueness metric based on canonical SMILES representations. Since evaluating molecular similarity in 3D space involves complex conformational and spatial considerations, we project molecules into 2D using canonical SMILES, a standardized molecular representation. Uniqueness is computed as the ratio of unique SMILES to the total number of valid SMILES for generated PROTAC molecules. This metric ranges from 0 to 1, where 1 indicates complete uniqueness, while lower values suggest structural repetition in the output.

### E.3.3. NOVELTY

Since the PROTAC-DB dataset represents only a tiny subset of the whole chemical space of PROTAC molecules, a model for PROTAC linker design with good coverage of chemical space will rarely generate PROTAC molecules present in the fine-tuning dataset. The metric of novelty assesses the fraction of generated PROTAC molecules that do not appear in the seen dataset (Brown et al., 2019). A higher novelty score indicates that the model can generate new, previously unseen PROTAC molecules, which is crucial for discovering new compounds. Therefore, models overfitting the dataset will often obtain low scores on this metric.

### E.3.4. RECOVERY

To evaluate a model's ability to recreate existing PROTAC linkers, we use a recovery metric based on canonical SMILES matching. This metric quantifies the model's capacity to generate molecules that are identical to known, validated structures in the training set, providing insight into the model's ability to capture the essential features of successful PROTAC linkers. The metric recovery is defined as the ratio between matched structures and total generated molecules. A recovery ratio that is too low may indicate insufficient learning of essential structural features, while an exceptionally high ratio might suggest overfitting to known structures rather than meaningful generalization.

### E.3.5. QUANTITATIVE ESTIMATE OF DRUG-LIKENESS (QED)

The Quantitative Estimate of Drug-likeness (QED) (Bickerton et al., 2012) serves as a comprehensive metric for assessing the drug-like characteristics of the molecules. QED integrates eight physicochemical parameters: molecular weight, octanol-water partition coefficient (logP), rotatable bonds, hydrogen bond donors, hydrogen bond acceptors, polar surface area, aromatic rings, and the number of structural alerts. These parameters are weighted and combined through a series of desirability functions derived from the analysis of known oral drugs, resulting in a continuous score between 0 and 1, where higher values indicate more favorable drug-like properties.

### E.3.6. SYNTHETIC ACCESSIBILITY (SA)

The Synthetic Accessibility (SA) Score aims to assess the synthetic difficulty of chemical compounds (Ertl & Schuffenhauer, 2009). The score combines multiple molecular features such as ring complexity, stereochemical complexity, structural symmetry, and the presence of uncommon structural elements to estimate the synthetic feasibility of a target compound. Typically scaled from 1 to 10, where lower values indicate easier synthesis, the SA score is rule-based and penalizes the occurrence of fragments rarely found in a reference dataset and the presence of specific structural features.

### E.3.7. MINIMUM ENERGY $E_{\text{MIN}}$

To quantitatively assess the quality of 3D conformations, we perform energy minimization using the Merck Molecular Force Field (Halgren, 1996). For each generated PROTAC molecule with $N$ conformers, we define $E_{\text{min}}$ as the average minimum energy of generated molecules per fragment pair before optimization. A lower $E_{\text{min}}$ indicates the better quality of the overall generated conformations.

### E.3.8. ROOT MEAN SQUARE DEVIATION (RMSD)

The Root Mean Square Deviation (RMSD) is the standard metric used to quantify the conformational similarity between pre-optimization and post-optimization structures, indicating how much the atomic positions of a molecule have shifted during the optimization process, with a lower RMSD signifying greater similarity between the two structures.

### E.4. Implementation Details

We view the input fragments and linkers as fully connected graphs. The atom features include a one-hot element and a charge indicator including H, C, N, N⁻, N⁺, O, O⁻, F, Cl, Br, I, S(2), S(4), and S(6). We also add one one-hot value to indicate if the molecular graph is a fragment or a linker. Similarly, the edge features include a one-hot bond type indicator (None, Single, Double, Triple, Aromatic), and a 4-dim one hot vector indicating the edge is between fragment atoms, linker atoms, fragment-linker atoms, or linker-fragment atoms.

During the pre-training phase, atom features and edge features are first fed into two MLP embedding layers with a dimension of 256 for nodes and a dimension of 64 for edges. These hidden embeddings get involved in an atom update layer, bond update layer, and position update layer in the EGNN model. In each EGNN layer, we concatenate the input features and update the hidden embeddings or positions with a 2-layer MLP with LayerNorm and ReLU activation function. We stack 3 layers of EGNN for denoising. We set the number of diffusion steps as 500. For the diffusion noise schedule, we choose a cosine $\beta$ schedule suggested in (Nichol & Dhariwal, 2021) with $s = 0.01$. The model is trained via AdamW (Loshchilov & Hutter, 2019) with the initial learning rate 5e-5, $\beta_1 = 0.99$, $\beta_2 = 0.999$. We also schedule to decay the learning rate exponentially with a factor of 0.6 and a minimum learning rate of 1e-6. During the fine-tuning phase, we also adopt a 4-layer MLP with 512 hidden units and SiLU activation function as the backbone of the guidance network. We train the guidance network for 60k iterations by Adam optimizer with a learning rate of 3e-4.

## F. Additional Experimental Results

### F.1. Distributions of Molecular Descriptors of the Two Datasets

We further illustrate the distributional differences between small molecules and PROTACs in chemical space. In Figure 8, we highlight additional key physicochemical distinctions, including partition coefficient (LogP), rotatable bond count (flexibility and conformational dynamics), and the number of hydrogen bond donors (HBDs) and acceptors (HBAs). PROTACs are sourced from PROTAC-DB and PROTACpedia [1], while small molecules are randomly sampled from ZINC.

### F.1.1. MOLECULAR WEIGHT

As shown in Figure 2, small molecules typically range between 250–500 Da, a range optimized for oral bioavailability according to Lipinski's Rule of Five. In contrast, PROTACs peak at 750–1000 Da, indicating their significantly larger size. The small-molecule MW distribution is narrow and sharply peaked, reflecting a relatively uniform size range, whereas the broader PROTAC distribution indicates greater structural variability.

---

[1] https://protacpedia.weizmann.ac.il/ptcb/main

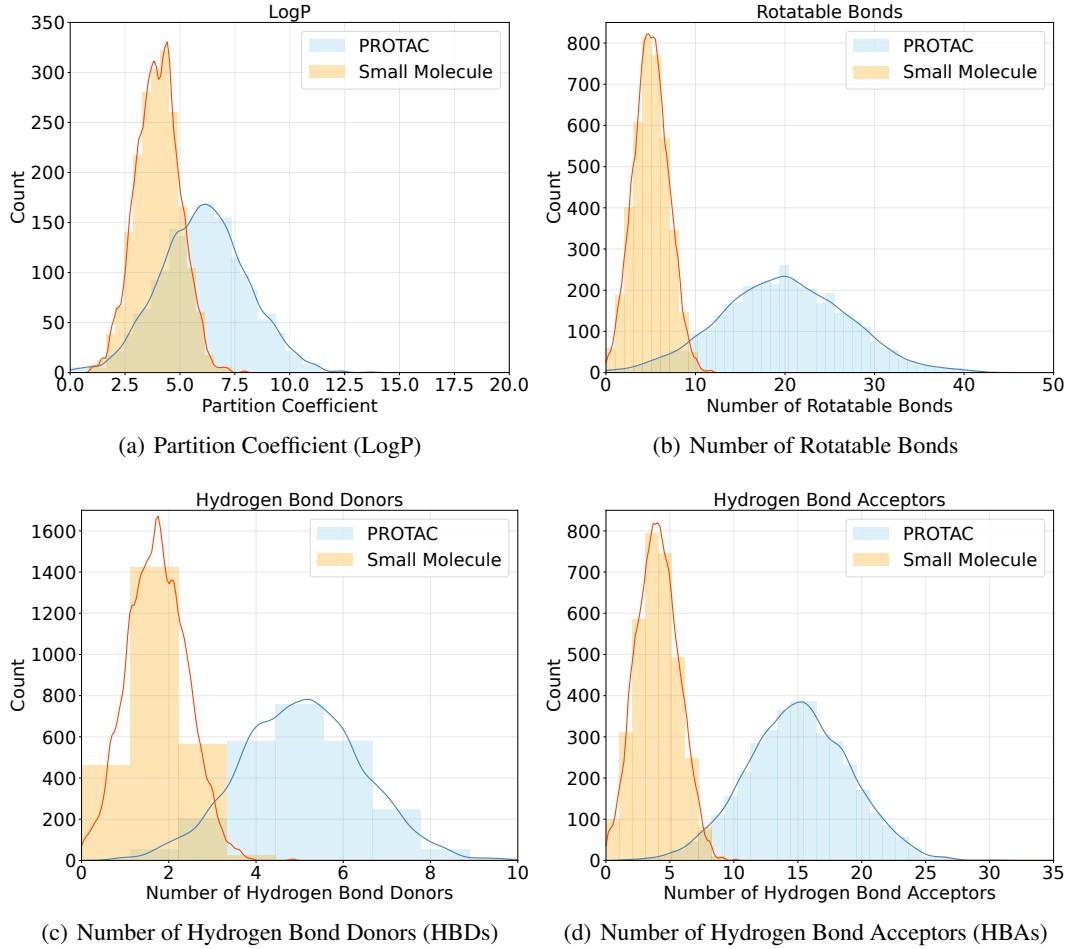

Figure 8: The distributions of various molecular descriptors of PROTACs versus small molecules in the chemical space.

### F.1.2. PARTITION COEFFICIENT (LOGP)

The LogP distribution for small molecules peaks around 3-4, aligning with drug-likeness criteria that favor values between 1 and 3 for optimal oral bioavailability. PROTACs, however, exhibit a broader LogP distribution, peaking around 6, indicating higher hydrophobicity. This increased hydrophobicity can impact solubility and cellular permeability, posing challenges for aqueous environments such as extracellular fluids and the cytosol.

### F.1.3. ROTATABLE BONDS

Small molecules typically have 1–10 rotatable bonds, contributing to their relatively rigid structures. In contrast, PROTACs peak at 10–20 rotatable bonds, largely due to their flexible linker regions.

### F.1.4. HYDROGEN BONDING

The HBD distribution for small molecules peaks around 1–2, consistent with drug-likeness criteria that prioritize membrane permeability. PROTACs, however, peak around 5 HBDs, with a similar trend observed for HBAs. These differences further distinguish PROTACs from traditional small molecules in terms of molecular interactions.

Thus, models trained on small molecule datasets like ZINC fail to capture the distinct characteristics of PROTACs, as small molecule fragments and PROTACs occupy largely non-overlapping regions of chemical space. This motivates the development of a domain-adapted generative model that explicitly accounts for distributional differences between the small molecule and PROTAC domains.

