# OpenReview forum: "Domain-Adapted Diffusion Model for PROTAC Linker Design Through the Lens of Density Ratio in Chemical Space"
_ICML.cc/2025/Conference — ICML 2025 poster_

### Official Review · Reviewer_omEm · 2025-03-10

**Overall Recommendation:** 3

**Summary:**

In this work the authors explore a domain-adapted diffusion model for unconditional molecular generation, focusing on PROTAC linker design. While this is an interesting application, the novelty is somewhat limited as it does not explore one of the most compelling aspects of molecular design: conditional molecular generation. The challenge of generating molecules of this size in an unconditional setting is largely addressed in existing work, unless specific metrics to assess chemical space exploration were included (which are not present here). While the background and baseline choices are relevant, they do not introduce particularly novel insights.

Overall, this paper is close to the acceptance threshold, and with additional experiments and analysis, it could become more impactful for both the machine learning and molecular engineering communities, but I have ranked it as a weak reject in its current state.

Summary of Feedback and Concerns

1. Reproducibility Concerns:

    - Unclear how training, validation, and/or test sets were determined (limited details in paper, and no code provided).

    - Lack of clarity on structures sampled from ZINC for pre-training (were they quasi-PROTACs?); furthermore, unclear why the molecular splitting strategy that the authors used for pre-training makes sense, was it just to generate as many splits as possible for the quasi-PROTAC pre-training data?

    - How were SMILES, and eventually conformers, in the pre-training and fine-tuning sets standardized and processed prior to training? There is a lot of ambiguity in data processing and standardization steps that is not clarified in the text, yet this is really important.

    - The statement regarding PROTAC data for fine-tuning is confusing: _"We select 365 different warheads as the test set of 327 PROTAC samples, and the remaining as the training set of 2,943 samples for the fine-tuning phase."_ Unclear what is meant by this and frankly the statement does not make much sense to me. Hope the authors can clarify.

    - No code availability, making it difficult (impossible) to validate and reproduce the work.

2. Model Training and Evaluation:

    - I am concerned over the computational cost of training and sampling the diffusion model, despite the authors' claim: _"fine-tuning approach in the DAD-PROTAC model enjoys two main advantages. First, it is computationally efficient through the score estimator correction rather than full model retraining. Second, it explicitly estimates the density ratio in the chemical space with theoretical rigor for effective domain adaptation."_ Could the authors provide some more details here about the computational cost for training and inference? For instance, how long does it take to generate a linker for a new structure?

    - The authors' benchmarking choices are relevant but could have been stronger, they compare against 3DLinker, DiffLinker, and LinkerNet; however, REINVENT (Link-INVENT) would have been a more suitable benchmark as it is one of the best molecular generative models publicly available, even though it does not explicitly use 3D data (see my concerns below about this).

3. Justification for the 3D Representation:

    - The authors should better justify why 3D structures are necessary for this task. It is unclear whether a 2D representation would have sufficed, and whether the complexity of using 3D models is warranted.


4. Benchmarking and Evaluation Limitations:

     - The model appears to be generating PROTAC linkers unconditionally, rather than for specific POIs (proteins of interest) and E3 ligases, limiting its practical utility. Future work should focus on generating PROTACs for specific targets and evaluating performance across different targets.

    - The focus on basic molecular generation benchmarks (e.g., validity) is not very insightful, as any standard generative model should meet these. Rediscovery benchmarks would have been more meaningful to demonstrate the model’s practical applicability for PROTAC linker design; however, these are only really feasible in a conditional setting, which the authors did not explore.

    - Sample efficiency should be evaluated, as PROTAC-based generative models must be sample-efficient due to the limited amount of publicly available PROTAC data. Was this explored at all?

5. Overall Summary:

    - None of the presented experiments convincingly demonstrate that this model would be useful in a prospective setting; as such, I question the utility of it. The use or need for 3D data in this setting was not really motivated, so I think the authors need to come back to this. Currently, the study feels incomplete, unless the aims of the study are reframed from PROTAC design to simply domain-adaptation. IMO, sample efficiency and rediscovery benchmarks should be prioritized to establish the model’s utility for PROTAC design.

**Claims And Evidence:**

Partially. See my detailed review above.

**Essential References Not Discussed:**

Perhaps Link-INVENT https://doi.org/10.1039/D2DD00115B

**Experimental Designs Or Analyses:**

Analysis is incomplete. See my detailed review above.

**Methods And Evaluation Criteria:**

Partially. See my detailed review above.

**Other Comments Or Suggestions:**

Overall it is not a bad paper, I want to make that clear, I quite enjoyed reading it. Nevertheless, it can be improved through the analysis and experiments recommended above.

**Other Strengths And Weaknesses:**

The paper is overly convoluted, with a heavy focus on the math and derivations for the domain-adaptation via the density ratio, without a strong-enough emphasis on if the evaluations are the most relevant for the model. Overall the paper could be restructured for clarity.

**Questions For Authors:**

See my detailed review above.

**Relation To Broader Scientific Literature:**

Decent review of the broader scientific literature.

**Theoretical Claims:**

Fine.

---

> ### Author Rebuttal · Authors · 2025-03-31
>
> Thanks for the constructive comments!
>
> # 1. Experimental Details
> Regarding the training/test split, since the proposed model is a pretrain-finetuning model, we use different datasets for the two phases. We use the ZINC dataset (438610 small molecule samples as the training set) to pretrain the model (Sec. 3.1.1, lines 283-293 and Appendix E.1.1). We use the PROTAC-DB dataset during the fine-tuning phase. More specifically, 2943 PROTAC samples of the PROTAC-DB dataset are used for fine-tuning (including both the training and validation sets), while the remaining 327 PROTAC samples of the PROTAC-DB dataset are used as the test set (Sec. 3.1.1, lines 295-300 and Appendix E.1.2).
>
> Regarding the structures sampled from ZINC, they can be viewed as quasi-PROTACs. We follow the existing molecular splitting method on the ZINC dataset for the linker design task in DiffLinker[1]. The molecules are fragmented by enumerating all double cuts of acyclic single bonds that are not within functional groups. The resulting splits are filtered by SA, PAINS, and other criteria. Please refer to Appendix E.1.1.
>
> Regarding the data pre-processing, we do not use SMILES since we focus on 3D structure generation. The input should be conformers. We first generate 3D conformers using RDKit and define a reference structure for each molecule by selecting the lowest-energy conformations. This also follows the practice in [1]. Please refer to Appendix E.1.1.
>
> Regarding the PROTAC data for fine-tuning, we split the PROTAC dataset into training (2943 samples) and test (327 samples) sets. The **test set contains 365 unique warhead pairs not seen in the training set** to evaluate generalization. The mismatch in numbers (327 PROTACs vs. 365 warheads) arises since some warheads are shared across PROTACs. We will revise it for clarity.
>
> Regarding code, please refer to Appendix E.4, end of page 24.
>
> [1] Ilia Igashov et al., Equivariant 3D-conditional diffusion model for molecular linker design.
>
> # 2. Model Training and Evaluation
> Regarding the computational cost for training and inference, please refer to Appendix E.4, lines 1305-1308. Our model can converge within 77 hours during the pre-training phase. For the fine-tuning phase, the guidance network can converge within 31 hours. For the sampling efficiency, DAD-PROTAC can sample one PROTAC linker within 29 minutes. It is efficient compared to other baselines in Figure 4(a).
>
> Regarding the Link-INVENT, we do not include it as a baseline since it does not use the 3D molecular graphs as the input. However, we can transform the 3D graphs in the datasets to their corresponding SMILES strings or 2D graphs, and then use SMILES/2D graphs to pretrain and fine-tune the Link-INVENT. We then compare it with DAD-PROTAC below. The Link-INVENT performs well on the valid metric but performs quite badly on metrics like uniqueness and novelty. This is because it does not use 3D information and can only generate the linker in the restrained 2D space, making it difficult to generate unique or novel structures. **We will cite the Link-INVENT paper[2] and compare its performance with ours.**
>
> | Models | Valid% | Unique% | Novel% | Recover% |
> |:---:|:---:|:---:|:---:|:---:|
> | 3DLinker-Fine-tuning | 58.6 ± 0.2 | 49.2 ± 0.6 | 56.2 ± 0.5 | 24.3 ± 0.4 |
> | LinkerNet-Fine-tuning | 82.9 ± 0.3 | 54.6 ± 7.0 | 63.7 ± 3.8 | 32.8 ± 0.8 |
> | Link-INVENT-Fine-tuning | 93.6 ± 0.5 | 43.7 ± 0.9 | 52.4 ± 1.4 | 29.1 ± 1.7 |
> | DAD-PROTAC | **94.8 ± 0.4** | **69.3 ± 0.3** | **71.5 ± 0.3** | **45.7 ± 0.6** |
>
> [2] Jeff Guo et al., Link-INVENT: generative linker design with reinforcement learning. Digital Discovery, 2023.
>
> # 3. Justification for the 3D Representation
> If we only use SMILES or 2D graphs, atoms in the 3D space that are close in the molecule structure can be far away in the SMILES string/ 2D graphs. The 3D representation contains information about relative distance and orientation between the sub-structures, which is vital to successful PROTAC linker design[3]. While 2D representations capture connectivity, they ignore torsional flexibility and spatial constraints critical for PROTAC efficacy. Most SOTA methods(Delinker, DiffLinker, LinkerNet) all use 3D representations. We will add more discussions to the paper. Please check the comparison with Link-INVENT above.
>
> [3] Fergus Imrie et al., Deep Generative Models for 3D Linker Design. Journal of Chemical Information and Modeling, 2020.
>
> # 4. Benchmarking
> Regarding conditional PROTAC linker generation, we leave it for future work (Sec. 4, lines 435-438).
>
> Regarding the rediscovery benchmark, we include the recovery rate (Appendix E.3.4) as the evaluation metric in Tables 1 and 2.
>
> Regarding sample efficiency, please refer to **the response to Reviewer 3rte: Performance with different sizes of PROTAC datasets**.
>
> # End of Response
> We hope that our responses have effectively addressed all of your concerns. If so, we would appreciate it if you could increase your rating accordingly.

---

> > ### Comment · Reviewer_omEm · 2025-04-03
> >
> > Thank you to the authors for the thoughtful response, as well as the additional data preparation details and the comparison to LinkINVENT. I have found it very informative, and believe these results should also be incorporated into the manuscript. I will update my score to a 3 if these components are suitably integrated in the revised manuscript, as well as the notes below regarding the code.
> >
> > Apologies for missing the anonimized code repository, it was an oversight on my part. It is very good that it has been provided, however, the documentation is limited: there are not really set-up instructions in the README nor any example use cases, which limits its utility and makes it hard to follow. This is especially problematic since a lot of these details essential for reproducibility have not been present in the initial draft of the manuscript. Can the authors please integrate better documentation and improve the usability in the revised code repository, along with the revised manuscript? (to make it unambigious: I would also like to see an improved repository in order to increase my score to a 3)
> >
> > Finally, I am not convinced by the aforementioned justification that the 3D representation is needed, as transformers are also able to capture long-range dependencies from 2D representations like SMILES. Indeed, a more thorough discussion on the strengths and limitations of the chosen representation will improve the paper.

---

> > > ### Author Response · Authors · 2025-04-04
> > >
> > > Thanks so much for your further insightful and valuable comments! Your suggestions really make this work better. Rest assured that we will definitely revise the manuscript in the final version to incorporate all your informative suggestions, especially the additional data preparation details and the comparison to LinkINVENT.
> > >
> > > ---
> > > Regarding repository documentation, we have improved the README file significantly. Please still refer to Appendix E.4, end of page 24. We now include the step-by-step running instructions, including a detailed setup guide for environment configuration, how to do data preprocessing, how to pretrain the model, how to fine-tune the model, how to sample the linkers, and how to evaluate the performance of the sampled results. We also provide some example data files and the pre-trained model and fine-tuned model weights as well. The workflow of this repository is now clear.
> > >
> > > ---
> > > Regarding justification for the 3D representation, we will expand our discussion on the strengths and limitations of 3D representation in the final revised manuscript. While it is true that transformers can capture long-range dependencies in SMILES or 2D representations, we emphasize that the **3D representation encodes explicit geometric information** about inter-atomic distances, orientations, and torsional angles. **Such spatial and conformational details are missing in SMILES or 2D representation, but they are vital to successful PROTAC linker design.** Our comparison with LinkINVENT and the results in [1] both empirically show that omitting such spatial and conformational information harms the performance of the linker design task. We will definitely include these comparison results with LinkINVENT to support this point.
> > >
> > > [1] Fergus Imrie et al., Deep Generative Models for 3D Linker Design. Journal of Chemical Information and Modeling, 2020.
> > >
> > > ---
> > > We believe these improvements comprehensively address all the concerns while strengthening the overall paper. If so, we would appreciate it if you could increase your rating accordingly. Thanks again so much for your great suggestions!

---

### Official Review · Reviewer_3rte · 2025-03-14

**Overall Recommendation:** 4

**Summary:**

This paper introduces DAD-PROTAC, a domain-adapted diffusion model for designing linkers in Proteolysis-targeting chimeras (PROTACs). The main algorithmic idea is the efficient fine-tuning strategy via density ratio estimation, avoiding full retraining of the diffusion model.

**Claims And Evidence:**

The claims made in the submission are generally supported by clear and convincing evidence:

* **Claim:** Existing diffusion models for linker design, trained on small-molecule datasets, suffer from a distribution mismatch when applied to PROTACs.

  **Evidence:** Figure 2 clearly shows the difference in molecular weight distributions. Appendix B.1 and Figure 7 and F.1, along with the accompanying text, provides substantial additional detail, discussing differences in data collection and various physicochemical properties (LogP, rotatable bonds, etc.).

* **Claim:** DAD-PROTAC's domain adaptation, via density ratio estimation, improves performance.

  **Evidence:** Table 1 shows superior performance of DAD-PROTAC compared to baselines (3DLinker, DiffLinker, LinkerNet) and their fine-tuned versions across multiple metrics (validity, uniqueness, novelty, recovery, QED, SA, Emin, RMSD). Table 2 presents a convincing ablation study, demonstrating the importance of the density ratio estimation and the use of noise-perturbed samples.

* **Claim:** DAD-PROTAC is more computationally efficient than full fine-tuning.

  **Evidence:** Figure 4(a) shows significantly reduced fine-tuning time compared to standard fine-tuning approaches, while achieving higher validity. The text discusses the computational cost savings of learning a classifier for the score correction term rather than retraining the entire model.

* **Claim:** The generated linkers are closer to real protacs.

  **Evidence:** Figure 5 gives the distribution of molecular weight of the generated linkers.

Overall, the claims are well-supported by a combination of quantitative results, qualitative visualizations (Figures 1, 5, and Appendix figures), and theoretical justification (Theorems 2.1 and 2.2).

**Essential References Not Discussed:**

No essential references appear to be missing.

**Experimental Designs Or Analyses:**

I checked the soundness/validity of the experimental designs and analyses. The ablation study (Table 2) is particularly well-designed, isolating the contributions of different components of the proposed method. The comparisons to baselines are fair, with pre-trained versions of the baselines included. The use of multiple metrics provides a comprehensive evaluation. The visualization results in Figure 5 support the claims.

**Methods And Evaluation Criteria:**

The proposed methods and evaluation criteria are appropriate for the problem.

* **Methods:** The core idea of using a diffusion model with domain adaptation via density ratio estimation is well-motivated and theoretically grounded. The decomposition of the score estimator is a clever way to leverage pre-trained models while adapting to the target domain. The use of EGNNs is standard and appropriate for handling 3D molecular structures.

* **Evaluation Criteria:** The paper uses a comprehensive set of metrics relevant to molecular generation:

* **Validity, Uniqueness, Novelty:** Standard metrics for assessing the quality and diversity of generated molecules.

* **Recovery:** Measures the ability to reproduce known linkers, which is important for validating the model's ability to learn the underlying distribution.

* **QED, SA:** Established metrics for assessing drug-likeness and synthetic accessibility.

* **Emin, RMSD:** 3D conformation-specific metrics that evaluate the quality of the generated structures.

* **Datasets:** Using ZINC for pre-training and PROTAC-DB for fine-tuning and evaluation is a reasonable choice, reflecting the availability of data in these domains. The paper clearly explains the data processing and splitting procedures.

* **Baselines:** The comparison to 3DLinker, DiffLinker, and LinkerNet (and their fine-tuned versions) provides a good assessment of the proposed method's performance relative to existing state-of-the-art approaches.

**Other Comments Or Suggestions:**

Reference the **Other Strengths And Weaknesses** part

**Other Strengths And Weaknesses:**

**Strengths:**

* **Novelty:** The core idea of using density ratio estimation for domain adaptation in PROTAC linker design is novel and well-motivated.
*  **Thoroughness:** The paper is very thorough, with extensive experiments, ablations, visualizations, and theoretical justifications.
* **Clarity:** The paper is generally well-written and easy to follow. The figures and tables are informative and well-designed.

**Weaknesses:**

* The Figure 3 of DAD-PROTAC model is a little complicated, and is hard to understand.

**Questions For Authors:**

1. The current approach assumes the number of atoms in the linker is pre-specified. How challenging would it be to extend the model to generate linkers of variable lengths? How would this impact the density ratio estimation and score correction steps? This would significantly improve its real applicability.

2. How does DAD-PROTAC's performance scale with the size of the PROTAC training dataset? The paper mentions the limited size of PROTAC datasets as a challenge. Could you provide some insights into how the method would perform with even smaller or larger datasets? A learning curve would help in assesing this limitation.

**Relation To Broader Scientific Literature:**

The paper is well-situated within the broader scientific literature. It clearly explains the context of PROTAC linker design and the limitations of existing approaches. The related work section (and Appendix A) provides a good overview of relevant work in molecular generation, diffusion models, and PROTAC design. The paper cites relevant papers on diffusion models (Hoogeboom et al., 2022; Nichol & Dhariwal, 2021), EGNNs (Satorras et al., 2021), molecular generation (Guan et al., 2023; Igashov et al., 2024), and PROTACs (Bemis et al., 2021; Troup et al., 2020).

**Theoretical Claims:**

I checked the correctness of the proofs for theoretical claims.

Theorem 2.1 is correct. The decomposition is valid.

Theorem 2.2 is correct. The training objective is derived in the appendix.

The demonstrations are given in Appendix C and B.4, and they are corrects.

---

> ### Author Rebuttal · Authors · 2025-03-31
>
> Thank you for the constructive comments!
>
> # 1. Figure 3
>
> We will make Figure 3 clearer in the final version with fewer annotations and highlight more on how the score correction term is obtained. We will also explicitly annotate the input and output of two phases and each model component.
>
> # 2. Pre-specified number of atoms
> We **follow most existing SOTA linker design methods like LinkerNet[1] and also pre-specify the number of atoms**. This is because the prediction of the number of atoms in the linker should be solved via **a deterministic model**, while the generation of the linker (atom coordinates and bonds) should be solved via another **generative model**. **The two steps of the linker design task are usually solved separately**. We focus on the generation part of this task in this work.
>
> For the prediction of the number of atoms in the linker, **we can use a separately trained GNN to produce probabilities for the linker size**. The input is still the molecule fragments. Later, when we ​​generate the linker’s structure, we first pre-specify the linker size based on the maximum predicted probabilities by this separately trained GNN and then use our proposed method in this paper. Since the linker size prediction step and the linker structure generation step are trained separately, **no components (density ratio estimation and score correction) in the proposed DAD-PROTAC would be affected**. We include the experimental results below. As we can see, **the additional training of another neural network for linker size prediction may introduce more errors for the linker size, and thus the performance may degrade**. We leave the joint training of both linker size prediction and linker structure generation for future work, as it is currently out of the scope of this work.
>
> |  | Valid% | Unique% | Novel% | Recover% |
> |:---:|:---:|:---:|:---:|:---:|
> | DAD-PROTAC w/o pre-specified number of atoms  | 93.1 ± 0.6 | 66.2 ± 0.7 | 69.3 ± 0.8 | 40.4 ± 0.5 |
> | DAD-PROTAC (Ours) | 94.8 ± 0.4 | 69.3 ± 0.3 | 71.5 ± 0.3 | 45.7 ± 0.6 |
>
> [1] Jiaqi Guan et al., LinkerNet: Fragment Poses and Linker Co-Design with 3D Equivariant Diffusion. In NeurIPS 2023.
>
> # 3. Performance with different sizes of PROTAC datasets
> **For sample efficiency**, we analyze the performance of the proposed DAD-PROTAC with different portions of the PROTAC dataset used during the fine-tuning phase in the table below. We reduce the PROTAC dataset size from 100% (full) to 50%, 25%, and 10%. We also add an extra 20% of samples from the PROTAC-DB 2.0 database as the extended dataset. This non-linear degradation suggests our approach is reasonably robust to limited PROTAC data for fine-tuning. Note that, **even at 10% of the PROTAC dataset size, DAD-PROTAC can still outperform the baseline 3DLinker-Fine-tuning approach using 100% of the PROTAC data**. For larger datasets, our preliminary experiments with additional data show diminishing returns with 1.2x our current dataset size, suggesting we are approaching the performance ceiling of the current architecture. **We will add these results and include a learning curve figure in the final version that visualizes these results for sample efficiency analysis.**
>
> | DAD-PROTAC with different sizes of datasets | Valid% | Unique% | Novel% | Recover% |
> |:---:|:---:|:---:|:---:|:---:|
> | 10% PROTAC dataset | 64.9 ± 0.8 | 56.8 ± 0.7 | 63.8 ± 0.5 | 31.4 ± 0.8 |
> | 25% PROTAC dataset | 76.3 ± 0.6 | 59.5 ± 0.5 | 67.9 ± 0.4 | 38.2 ± 0.7 |
> | 50% PROTAC dataset | 89.9 ± 0.5 | 65.8 ± 0.4 | 70.7 ± 0.3 | 43.1 ± 0.6 |
> | 100% PROTAC dataset | 94.8 ± 0.4 | **69.3 ± 0.3** | 71.5 ± 0.3 | **45.7 ± 0.6** |
> | 100% PROTAC dataset (full) + 20% additional | **95.1 ± 0.3** | 68.1 ± 0.3 | **71.9 ± 0.3** | 45.2 ± 0.5 |
>
> # End of Response
> We hope that our responses have effectively addressed all of your concerns.

---

> > ### Comment · Reviewer_3rte · 2025-04-08
> >
> > Thank you for your detailed rebuttal and for providing additional experiments and clarifications.
> > I appreciate you addressing my questions regarding:
> > 1.	Figure 3: Your plan to clarify Figure 3 in the final version is welcome.
> > 2.	Variable Linker Length (Q1): Thank you for the explanation regarding the standard practice of pre-specifying the linker length and for conducting the experiment with a separate length prediction step. The results provide valuable context and appropriately frame joint training as future work.
> > 3.	Dataset Size Sensitivity (Q2): The new experimental results analyzing performance with varying PROTAC dataset sizes are very helpful. They effectively demonstrate the robustness of DAD-PROTAC, even with smaller datasets, and provide useful insights into sample efficiency. Adding the learning curve figure will be a good addition.
> > Your responses and the new empirical evidence have successfully addressed the points raised in my review. My overall positive assessment of the paper remains, and I confirm my recommendation

---

### Official Review · Reviewer_gqa6 · 2025-03-14

**Overall Recommendation:** 3

**Summary:**

This study focuses on domain adaptation in diffusion models for biology. The authors pretrain a diffusion model on the ZINC dataset and attempt to use the model on the PROTAC domain. For finetuning, they use density ratio estimation techniques to correct the score function on the ZINC dataset. The method looks intersting and the results are great.

## Update after rebuttal
I appreciate the authors' efforts to answer the questions, most of my concerns are resolved. Thank you.

**Claims And Evidence:**

Yes, the claims are supported by the experiments.

**Essential References Not Discussed:**

NA

**Experimental Designs Or Analyses:**

Yes, I think this study is solid.

**Methods And Evaluation Criteria:**

Yes, the method and evaluation make sense.

**Other Comments Or Suggestions:**

NA

**Other Strengths And Weaknesses:**

**Strengths**

1. The domain adaptation diffusion is novel and interesting. Score correction helps preserve the original power of the pretrained model on ZINC and prevents from overfitting on the PROTAC dataset.
2. The results are great and outperform other methods. The ablation study proves the usefulness of the method.



**Weaknesses**

1. Training a classifier on x(t) may introduce some errors. For x(t) when t is large (in the early stage of reverse diffusion), will the classifier be hard to train? Since the atom number is so different (see Figure 2), will the classifier learn to cheat by counting the number of atoms?

2. Since you firstly train a classifier, and then train another network with the classifier, will error propagation hurt the performance? A baseline can be use Monte Carlo estimation to directly estimate the value of the score correction in Theorem 2.1. In Table 2, you have ablated "direct score correction approximation" and "density ratio estimation via clean samples", can you provide more details?

**Questions For Authors:**

**Questions**
There are other methods for "finetuning" a pretrained diffusion model on a specific domain, such as LoRA in LLM and diffusion DPO in image generation. Is it possible that these techniques help prevent overfitting?

**Relation To Broader Scientific Literature:**

It would benefit the linker design area.

**Theoretical Claims:**

I checked the claims.

---

> ### Author Rebuttal · Authors · 2025-03-31
>
> Thank you for the constructive comments!
>
> # 1.  Errors for large $t$
> In Eq.(16), we train a time-dependent classifier $W(X_t^L,t)$ with samples from potentially all steps $t$ jointly, instead of training different classifiers $W_t(X_t^L)$ for each step $t$ individually. This means that the **training set has both easy samples (t is small) and hard samples (t is large)**. This will mitigate the issue of differentiating the samples from two domains when t is large. Besides, we can easily **add more samples (with larger $m$ and $n$) to the training set to further mitigate this issue**. We also compare the classification performance to show the performance of this trained classifier on different groups of $t$ (based on its range) on an additional test set below. This table shows that **the model’s performance drop for large $t$ is acceptable**. The further **incorporation of focal loss to Eq.(16) by putting more focus on hard samples ($t$ is large)  is not necessary**. We will add it to the ablation study in the final version.
>
> | test samples (range of t) | 0% to 25%T | 25%T to 50%T | 50%T to 75%T | 75%T to 100%T |
> |---|---|---|---|---|
> | classification accuracy (with Eq.(16))(Ours) | 97.5 | 96.8 | 97.1 | 96.7 |
> | classification accuracy (with Eq.(16) + focal loss) | 97.3 | 97.2 | 97.2 | 96.9 |
>
> # 2. Model cheats by counting the number of atoms
> The number of atoms is only one of the differences between PROTACs and small molecules. **Please refer to Figure 8 of the paper to see other differences, like LogP, which are more likely to be dependent on the distributions of atom coordinates. Therefore, we train the classifier on atom coordinates $X$ in Eq.(16) to encode all potential distribution differences in their chemical space**. We specifically design our classifier to prevent this shortcut. We use original features like $X$ as the input rather than global molecular properties like the number of atoms. We validate this by testing the classifier on a separate dataset containing molecular structures with the same atom counts but from different domains (PROTACs and small molecules). The results in the table below confirm that **classification accuracy on test samples with the same atom counts remains consistent with the overall results**. If we only use atom counts as input, the classification performance would drop significantly.
>
> |  | test samples (overall) | test samples (same atom counts) |
> |---|---|---|
> | classification accuracy (use all original features as input) (Ours) | 97.0 | 96.7 |
> | classification accuracy (use only atom counts as input) | 71.3 | 53.2 |
>
> # 3. Error propagation and Monte Carlo estimation
> From the results in the last point above, we can see that **the classification error in this step is quite small**, and this alleviates the error propagation issue in the first place. We further find that using MC simulation to estimate the score correction term directly may get slightly better performance, but it is **computationally prohibitive**. The extended ablation study from Table 2 is summarized below. “direct score correction approximation” refers to the model where we only run MC simulations on a few samples $X_t^L$ to obtain the corresponding score correction terms in Eq.(11) and then train another neural network to learn to approximate this score correction term. “Density ratio estimation via clean samples” refers to the model where we only use clean samples to train the classifier in Eq.(16) (namely, set t =0). More discussion is found in Sec 3.3 in the paper. From the results in the table below, **MC simulation suffers from high computational cost, our current design keeps a good balance between effectiveness and efficiency.**
>
> |  | Valid% | Unique% | Novel% | Recover% | Relative Running Time |
> |---|---|---|---|---|---|
> | DAD-PROTAC w/ direct score correction MC simulation | 95.1 ± 0.6 | 70.1 ± 0.5 | 70.3 ± 0.4 | 46.2 ± 0.4 | 131x |
> | DAD-PROTAC w/ direct score correction approximation | 81.9 ± 0.8 | 49.5 ± 0.4 | 59.7 ± 0.3 | 28.1 ± 0.7 | 0.8x |
> | DAD-PROTAC w/ density ratio estimation via clean samples | 90.0 ± 4.7 | 62.8 ± 6.1 | 69.3 ± 0.2 | 41.0 ± 0.6 | 0.95x |
> | DAD-PROTAC (Ours) | 94.8 ± 0.4 | 69.3 ± 0.3 | 71.5 ± 0.3 | 45.7 ± 0.6 | 1x |
>
> # 4. Other methods for fine-tuning diffusion models
> LoRA and DPO are promising for parameter-efficient fine-tuning. However, they assume the source and target domains share a similar latent structure, which is invalid here due to the significant chemical differences between small molecules and PROTACs (Figure 8 in the paper). Our method explicitly models the domain shift via density ratios estimation, which is specifically designed for the linker design task. That said, integrating LoRA-style methods into our framework (e.g., for the correction term) could further improve efficiency. We leave it as future work since it is out of the scope of this paper.
>
> # End of Response
> We hope that our responses have effectively addressed all of your concerns.

---

### Decision · Program_Chairs · 2025-05-01

**Decision:**

Accept (poster)

**Comment:**

Dear authors,

thank you for submitting your paper that proposed DAD-PROTAC method. The revewiers found the topic important and timely. Your proposed algorithm is a diffusion model that leverages density ratio estimation for effective fine-tune on molecular generative models on PROTAC linker design tasks.
Your paper addresses the domain shift between small-molecule data (e.g., ZINC) and PROTAC chemical space that contains much bigger molecules. By utilizing the correction of the score function you managed to avoid a full retraining of the generative model.

In numerical experiments, the reviewers noted that your trained model achieves strong performance across a variety of metrics and your method demonstrates sample efficiency, and is supported by thorough ablation studies.

I would like to strongly encourage to revise your paper for camera-ready version and incorporate all the clarifications that were prepared during the rebuttal phase.

Congratulation again,

best, AC